# CCL5 promotes breast cancer recurrence through macrophage recruitment in residual tumors

Andrea Walens*, Ashley V DiMarco, Ryan Lupo, Benjamin R Kroger, Jeffrey S Damrauer, James V Alvarez*

Department of Pharmacology and Cancer Biology, Duke University, Durham, United States

**Abstract** Over half of breast-cancer-related deaths are due to recurrence 5 or more years after initial diagnosis and treatment. This latency suggests that a population of residual tumor cells can survive treatment and persist in a dormant state for many years. The role of the microenvironment in regulating the survival and proliferation of residual cells following therapy remains unexplored. Using a conditional mouse model for Her2-driven breast cancer, we identify interactions between residual tumor cells and their microenvironment as critical for promoting tumor recurrence. Her2 downregulation leads to an inflammatory program driven by TNFα/NFκB signaling, which promotes immune cell infiltration in regressing and residual tumors. The cytokine CCL5 is elevated following Her2 downregulation and remains high in residual tumors. CCL5 promotes tumor recurrence by recruiting CCR5-expressing macrophages, which may contribute to collagen deposition in residual tumors. Blocking this TNFα-CCL5-macrophage axis may be efficacious in preventing breast cancer recurrence.
DOI: https://doi.org/10.7554/eLife.43653.001

## Introduction

In 2018, it is estimated that approximately 270,000 women will be diagnosed with breast cancer, and 41,000 women will succumb to the disease (*Siegel et al., 2018*). Historically, over half of these deaths are due to recurrence 5 or more years after initial diagnosis and treatment (*Sosa et al., 2014*). This suggests that in a subset of patients, there is a population of clinically undetectable residual tumor cells that survive therapy, and may serve as a reservoir for eventual relapse. The long latency of recurrence has led to speculation that residual tumor cells are slowly growing or even dormant (*Hölzel et al., 2010*; *Klein, 2009*). Understanding how residual cells survive therapy, persist in a non-proliferative state, and eventually resume proliferation to form recurrent tumors is critical for preventing recurrences.

Much of the work examining mechanisms of tumor cell survival and recurrence following therapy has focused on tumor cell-intrinsic pathways (*Sosa et al., 2011*). Genetic mutations that render cells resistant to therapy represent an important mechanism of survival (*Holohan et al., 2013*), but there is emerging evidence that non-genetic pathways can also promote survival in response to therapy. For instance, a population of cells called drug-tolerant persisters has been shown to survive therapy through epigenetic adaptations (*Sharma et al., 2010*). Additionally, epithelial-to-mesenchymal transition has been shown to promote cell survival in response to EGFR inhibitors (*Sequist et al., 2011*). Finally, alterations in apoptotic pathways within tumor cells can promote cell survival in response to both chemotherapy and targeted therapy (*Alvarez et al., 2013*; *Damrauer et al., 2018*; *Hata et al., 2016*; *Holohan et al., 2013*; *Mabe et al., 2018*). In spite of this extensive literature on cell-intrinsic mechanisms of therapeutic resistance, much less is known about tumor cell-extrinsic contributions to

*For correspondence:
andrea.walens@duke.edu (AW);
james.alvarez@duke.edu (JVA)

Competing interests: The authors declare that no competing interests exist.

**eLife digest** Breast cancer is the second-leading cause of cancer-related deaths in women. Recurrence of breast-cancer five or more years after initial diagnosis and treatment causes more than half of these deaths. This suggests that some tumor cells survived treatment and persisted undetected. These residual tumor cells may not grow for years and are often surrounded by other cells, including immune system cells. What role these surrounding immune cells play in triggering future growth of these residual tumor cells is not clear.

Many breast cancer patients receive chemotherapy, which kills all quickly dividing cells. Targeted therapies, which block signals necessary for cancer cell growth, are also used often. More recently, scientists have developed treatments that use a patients own immune system to fight off cancer. Scientists are currently studying whether combining these immunotherapies with chemotherapy or targeted therapies increases the likelihood of eliminating cancer. Learning more about the role surrounding immune cells play in allowing residual tumor cells to persist and regrow is important to understanding how to treat cancer more successfully and prevent recurrence.

Now, Walens et al. show that immune cells called macrophages supply residual breast cancer cells in mice with a protein called collagen that they need to grow. In the experiments, mice with an aggressive form of breast cancer called Her2 received targeted cancer therapy. After the treatment, tumor cells in the mice released small molecules called cytokines that attract immune system cells. Levels of one cytokine called CCL5 rose after treatment and remained high in residual tumors in the mice. The experiments also revealed that CCL5 levels were high in residual breast cancer tumors collected from women.

This shows that high levels of CCL5 appear to shorten the amount of time between tumor treatment and recurrence because CCL5 attracts macrophages that deposit collagen in the residual tumors. Scientists believe collagen promotes tumor growth because recurrent tumors have high levels of collagen and breast cancer patients with high levels of collagen in their tumors often have worse outcomes. Treatments that prevent or block the release of CCL5 or that stop macrophages from supplying the residual tumor cells with collagen may help prevent recurrence.
DOI: https://doi.org/10.7554/eLife.43653.002

cell survival following therapy. Specifically, while there has been some recent focus on how the tumor microenvironment can promote tumor cell survival in response to therapy (*Meads et al., 2009*), little is known about whether the microenvironment regulates tumor cell survival, dormancy, and eventual recurrence.

We used a conditional mouse model of Her2-driven breast cancer to examine interactions between tumor cells and their microenvironment during tumor dormancy and recurrence. In this model, administration of doxycycline (dox) to bitransgenic MMTV-rtTA;TetO-Her2/neu (MTB;TAN) mice leads to mammary gland-specific expression of epidermal growth factor receptor 2 (Her2) and the development of Her2-driven tumors. Removal of dox induces Her2 downregulation and tumor regression. However, a small population of residual tumor cells can survive and persist in a non-proliferative state (*Alvarez et al., 2013*; *Moody et al., 2002*). These cells eventually re-initiate proliferation to form recurrent tumors that are independent of Her2. Using this model, we sought to understand how the interplay between tumor cells and their microenvironment regulates residual cell survival and recurrence.

## Results

### Her2 downregulation induces an inflammatory gene expression program driven by the TNFα/IKK pathway

To understand how interactions between tumor cells and their environment change in response to therapy, we first examined gene expression changes following Her2 downregulation in Her2-driven tumor cells. Two independent cell lines derived from primary Her2-driven tumors (*Alvarez et al., 2013*; *Moody et al., 2002*) were cultured in the presence of dox to maintain Her2 expression, or removed from dox for 2 days to turn off Her2 expression. Changes in Her2 expression following dox

withdrawal were confirmed by qPCR analysis (*Figure 1—figure supplement 1A*). Changes in gene expression were measured by RNA sequencing. Her2 downregulation led to widespread changes in gene expression in both cell lines (*Figure 1A*). Gene set enrichment analysis showed that an E2F signature was the most highly enriched gene set in cells with Her2 signaling on (+dox; *Figure 1—figure supplement 1B*), consistent with previous literature and the observation that Her2 is required for the proliferation of these cells (*Lee et al., 2000*). Interestingly, the gene sets most significantly enriched in cells following Her2 downregulation (-dox) were an inflammatory gene signature and a TNFα/ NFκB gene signature (*Figure 1B*). These gene sets comprised genes encoding chemokines in the CCL family (CCL2, CCL5, and CCL20) and CXCL family (CXCL1, CXCL2, CXCL3, CXCL5, and CXCL10), proteins that mediate cell-cell interactions (TLR2, ICAM1, and CSF1) as well as signaling components of the NFκB pathway (NFKBIA and NFKBIE). All these genes were upregulated following Her2 downregulation (*Figure 1C*).

At high concentrations (>40 μg/ml) doxycycline itself can inhibit the NFκB pathway (*Alexander-Savino et al., 2016*; *Santa-Cecília et al., 2016*). Although the concentrations of dox (2 μg/ml) we use to culture primary tumor cells are well below these levels, we wanted to confirmed that the NFκB pathway activation observed following dox withdrawal was due to loss of Her2 signaling. To do this, we treated primary tumor cells with Neratinib, a small-molecule inhibitor of Her2, to inhibit Her2 signaling without removal of dox. Neratinib treatment led to an increase in phospho-p65 (*Figure 1—figure supplement 1C*), increased expression of TNFα (*Figure 1—figure supplement 1D*), and increased expression of the NFκB targets CXCL5 and CCL5 (*Figure 1—figure supplement 1E and F*). To further confirm that the low concentrations of dox used to culture primary tumor cells do not directly inhibit the NFκB pathway we treated NIH3T3 cells with TNFα in the presence or absence of 2 μg/ml dox and measured NFκB target genes. Dox treatment had no effect on the induction of NFκB target genes following TNFα treatment (*Figure 1—figure supplement 1G*). Taken together, these results demonstrate that Her2 inhibition leads to activation of the NFκB pathway.

Given the coordinated upregulation of these NFκB target genes, we reasoned that their expression may be induced by a common upstream secreted factor acting in an autocrine manner. To test this, we collected conditioned media from primary tumor cells grown in the absence of dox for 2 days. This conditioned media was supplemented with dox to maintain Her2 expression and added to naive primary tumor cells. Treatment with conditioned media led to a time-dependent upregulation of the pro-inflammatory chemokine CCL5 (*Figure 1D*). One common upstream mediator of this cytokine response is tumor necrosis factor alpha (TNFα), and we found that TNFα expression is increased between 10-fold and 100-fold following Her2 downregulation (*Figure 1E*). To test whether this is sufficient to activate downstream signaling pathways, we examined activation of the NFκB pathway following treatment with conditioned media from cells following Her2 downregulation. Indeed, we found that treatment of naive cells with Her2-off (–dox) conditioned media led to rapid, robust, and prolonged activation of the NFκB pathway as assessed by phosphorylation of p65 (*Figure 1F*). Importantly, Her2 levels remained high in these target cells (*Figure 1—figure supplement 1H*), indicating that Her2-off (–dox) conditioned media can activate the NFκB pathway even in the presence of Her2 signaling. In contrast, conditioned media from Her2-on (+dox) cells had no effect on p65 phosphorylation (*Figure 1—figure supplement 1I*). Finally, we tested whether the induction of chemokine genes following Her2 downregulation was dependent upon the NFκB pathway by treating cells with the IKK inhibitor, IKK16. We found that blocking IKK activity blunted the induction of all chemokine genes following dox withdrawal (*Figure 1G*). Taken together, these results suggest that Her2 downregulation leads to the induction of a pro-inflammatory gene expression program, likely driven by autocrine-acting TNFα and mediated through the IKK-NFκB pathway.

## Immune cell infiltration during tumor regression and residual disease

Her2 downregulation in Her2-driven tumors in vivo induces apoptosis and growth arrest, ultimately leading to tumor regression (*Moody et al., 2002*). However, a small population of tumor cells can survive Her2 downregulation and persist for up to 6 months before resuming growth to form recurrent tumors. These residual tumors can be identified histologically (*Figure 2A*). Many of the cytokines and chemokines induced shortly after Her2 downregulation function as chemoattractants for various immune cells (*Binnewies et al., 2018*; *López et al., 2017*). This led us to speculate that Her2 downregulation in vivo may promote infiltration of immune cells into the tumor. We therefore asked whether the immune cell composition of tumors changed during tumor regression and in residual

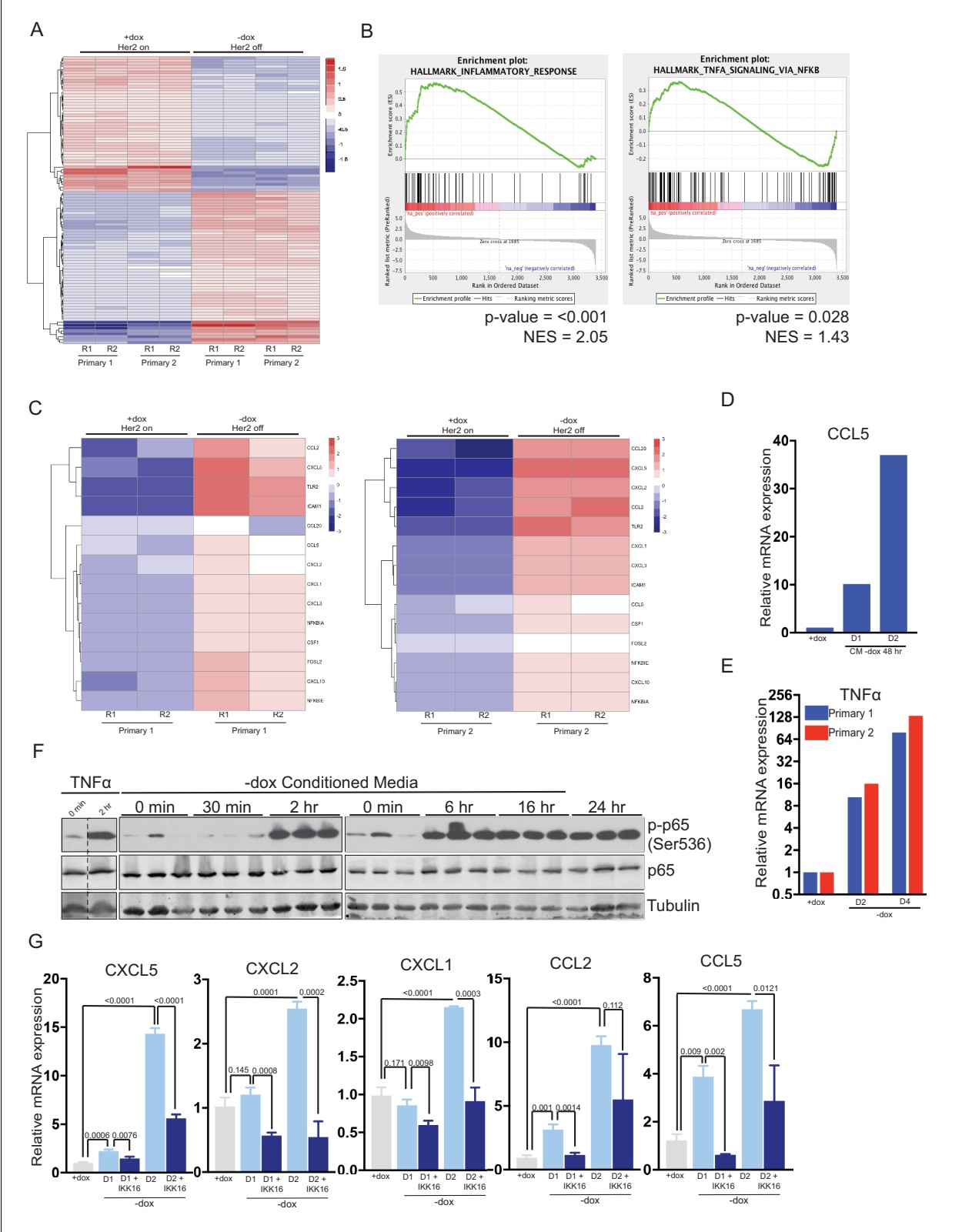

**Figure 1.** Her2 downregulation induces an inflammatory gene expression program driven by the TNFα/IKK pathway. (a) RNA-seq analysis of two independent primary Her2-driven tumor cell lines in the presence of Her2 expression (+dox) or 2 days following Her2 downregulation (-dox). The heatmap shows the top 100 differentially expressed genes between +dox and -dox conditions. R1 and R2 are biological replicates. (b) Gene set enrichment analysis (GSEA) of RNA-seq data showing enrichment of an inflammatory response signature and a TNFα/NF-κB signature in cells following

*Figure 1 continued on next page*

*Figure 1 continued*

Her2 downregulation. p-Values and normalized enrichment scores (NES) are shown. (c) Heatmap showing expression of select genes from the TNFα/ NF-κB signature in the presence of Her2 expression (+dox) or following Her2 deinduction (-dox). (d) qRT-PCR analysis of CCL5 expression following 1- or 2-day treatment with conditioned media harvested from primary cells following Her2 downregulation. Dox was added to conditioned media prior to treatment to maintain Her2 expression in target cells. Results shown are representative of two independent experiments. (e) qRT-PCR of TNFα expression in primary cells in the presence of Her2 expression (+dox) or 2 and 4 days following Her2 downregulation. Results shown are representative of two independent experiments. (f) Primary tumor cells were treated with conditioned media as described in (d), and activation of the NF-κB pathway was assessed by Western blot analysis of total and phospho-p65. Results show three biological replicates per time point. (g) qRT-PCR analysis of the indicated genes in primary tumor cells in the presence of Her2 expression (+dox) or 1 and 2 days following Her2 downregulation (-dox). At the time of Her2 downregulation, cells were treated with the pan-IKK inhibitor IKK16 (100 nM) or vehicle control. Results show the average of 3 biological replicates per condition. Error bars denote mean ± SEM. Significance was determined using a two-tailed Student's t-test.

DOI: https://doi.org/10.7554/eLife.43653.003

The following figure supplement is available for figure 1:

**Figure supplement 1.** Gene expression changes following Her2 inhibition.

DOI: https://doi.org/10.7554/eLife.43653.004

tumors. CD45 staining showed that leukocyte infiltration increased dramatically following Her2 downregulation as compared to primary tumors (*Figure 2B–C*, *Figure 2—figure supplement 1A*). Surprisingly, leukocytes remained high in residual tumors (*Figure 2D*, *Figure 2—figure supplement 1A*). Masson's trichrome staining revealed prominent collagen deposition in residual tumors (*Figure 2D*), consistent with a desmoplastic response in residual tumors. Staining for the macrophage marker F4/80 showed a dramatic increase in macrophage abundance during tumor regression (*Figure 2C*, *Figure 2—figure supplement 1A*), and macrophage levels remained elevated in residual tumors (*Figure 2D*, *Figure 2—figure supplement 1A*). CD3 staining showed increased T cell infiltration in regressing and residual tumors (*Figure 2—figure supplement 1A,B*). Taken together, these results indicate that Her2 downregulation leads to the infiltration of CD45+ leukocytes, and specifically F4/80+ macrophages. Residual tumors contain high numbers of macrophages and abundant collagen deposition, consistent with a desmoplastic response.

## Cytokine profiling of residual tumors

Immune cells can influence tumor cell survival and function (*Flores-Borja et al., 2016*; *Pollard, 2004*). The large number of immune cells present in residual tumors suggests that these cells may function to regulate the behavior of residual tumor cells. To begin to address this, we sought to identify secreted factors that are expressed in residual tumors. Residual tumor cells in the autochthonous MTB;TAN model are unlabeled and are diffusely scattered throughout the mammary gland, precluding their isolation. Therefore, we used an orthotopic model in which residual tumors can be easily isolated. In this model, primary Her2-driven tumors are digested, cultured, and infected with GFP. Cells are then injected into the mammary fat pad of recipient mice on dox to generate an orthotopic primary tumor. Following dox withdrawal, the fluorescently labeled residual tumors can be easily microdissected (*Figure 2—figure supplement 1C*). We first confirmed that the orthotopic model exhibited similar patterns of immune cell infiltration as the autochthonous model. Indeed, we found that macrophage staining increased dramatically during tumor regression and in residual tumors (*Figure 2—figure supplement 1D–F*), suggesting the orthotopic model is appropriate for identifying secreted proteins present in these residual tumors.

We generated a cohort of orthotopic primary tumors (n = 4) and residual tumors at 28 days (n = 6) and 56 days (n = 6) following dox withdrawal. Residual tumors were microdissected using a fluorescent dissecting microscope. We then made protein lysates from all samples and measured the expression of cytokines and chemokines using antibody-based protein arrays. Four primary tumors and four 28 day residual tumors were profiled using a commercially available cytokine array, which measures the expression of 20 secreted factors. We then used a second commercially available cytokine array, which measures 40 cytokines and chemokines, to measure cytokine expression in the whole cohort of tumors. This analysis identified eight cytokines that were upregulated in residual tumors as compared to primary tumors (*Figure 3A*; fold change >2, p < 0.1, *Figure 3—source data 1*), including CCL5, osteoprotegerin (OPG), and Vascular cell adhesion protein 1 (VCAM-1)

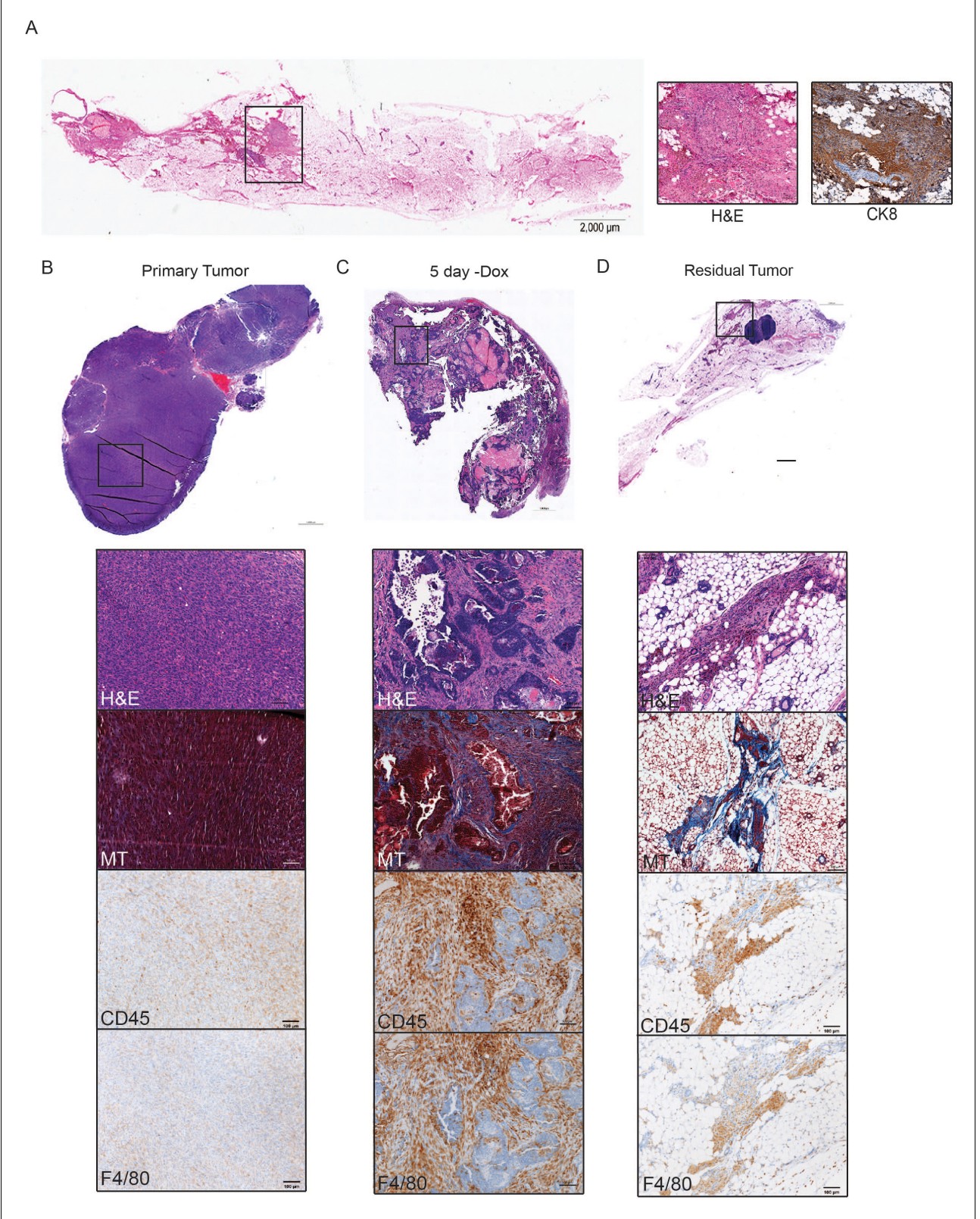

**Figure 2.** Immune cell infiltration during tumor regression and residual disease. (a) H and E-stained section of a representative residual tumor from a previously tumor-bearing MTB/TAN mouse. Insets show higher magnification view of residual tumor cells (left) and staining for CK8 (right). (b–d) Representative images of a primary tumor (b), regressing tumor (5 days -dox) (c), and residual tumor (d), stained with H and E, Masson's Trichome (MT),

*Figure 2 continued on next page*

*Figure 2 continued*

CD45, or F4/80. Primary tumors show little collagen deposition and only modest leukocyte infiltration. Her2 downregulation leads to infiltration of CD45 + cells, predominantly F4/80+ macrophages. Residual tumors have abundant collagen deposition and leukocyte infiltration.

DOI: https://doi.org/10.7554/eLife.43653.005

The following figure supplement is available for figure 2:

**Figure supplement 1.** Immune cell infiltration inautochthonousand orthotopic tumors following Her2 downregulation.

DOI: https://doi.org/10.7554/eLife.43653.006

(*Figure 3B*). Interestingly, VCAM-1 has been shown to regulate breast cancer dormancy (*Lu et al., 2011*), while OPG can regulate the survival of breast cancer cells (*Neville-Webbe et al., 2004*).

We next asked whether any cytokines were both induced acutely following Her2 downregulation and remained elevated in residual tumors. We found that only two cytokines, CCL5 and OPG, fulfilled these criteria. Given that OPG has previously been associated with dormancy, we focused our attention on CCL5. We then wanted to determine if CCL5 expression was elevated in human residual breast tumors following treatment. We analyzed a gene expression dataset of residual breast tumors that remain following neoadjuvant targeted therapy. A number of secreted factors were upregulated in residual tumors as compared to primary tumors, and CCL5 was one of the most significantly upregulated cytokines in this group (*Figure 3C–D* and *Figure 3—figure supplement 1A–M*). To confirm these results, we examined an independent gene expression data set from breast cancer patients treated with neoadjuvant chemotherapy. We found that CCL5 expression was also increased in residual tumors in this dataset (*Figure 3—figure supplement 1N*). These results suggest that CCL5 upregulation is a common feature of residual tumors cells that survive both conventional and targeted therapy in mice and humans, suggesting it may be functionally important in mediating the survival of these cells.

## CCL5 expression promotes recurrence following Her2 downregulation

We next wanted to directly assess whether CCL5 plays a functional role in regulating residual cell survival or recurrence. We first used an ELISA to measure CCL5 levels in orthotopic primary tumors, residual tumors, and recurrent tumors. CCL5 expression was elevated in residual tumors, confirming results from the cytokine array, and increased further in recurrent tumors (*Figure 4A*). We next engineered primary tumor cells to overexpress CCL5 or GFP as a control (*Figure 4B*) and used these cells in an orthotopic recurrence assay to test the effect of CCL5 expression on tumor recurrence. Control or CCL5-expressing cells were injected orthotopically into recipient mice on doxycycline to maintain Her2 expression. Primary tumors formed with similar kinetics following injection of control and CCL5-expressing cells, indicating that CCL5 expression had no effect on the growth of primary tumors (data not shown). Following primary tumor formation, mice were removed from dox to induce Her2 downregulation and tumor regression. Mice with residual tumors were palpated biweekly to monitor the formation of recurrent tumors. Tumors expressing CCL5 recurred significantly earlier than control tumors, indicating that CCL5 expression is sufficient to accelerate tumor recurrence (*Figure 4C*; p = 0.023; HR = 2.14).

We next asked if tumor-derived CCL5 is necessary for recurrence. To this end, we used CRISPR-Cas9 to knock out CCL5 in primary tumor cells (*Figure 4D*), and tested the effect of CCL5 knockout on recurrence using the orthotopic recurrence assay described above. The growth of CCL5 knockout tumors was not different from control tumors expressing a non-targeting sgRNA (data not shown). Mice were removed from dox, and the latency of recurrence between control and CCL5 knockout tumors was compared. We found that CCL5 knockout had no effect on the latency of recurrence (*Figure 4E*). Taken together, these results suggest that CCL5 expression is sufficient to accelerate recurrence, but tumor-derived CCL5 is not necessary for recurrence following Her2 downregulation.

## CCL5 promotes macrophage infiltration in residual tumors

CCL5 is a chemoattractant for various cell types, including T cells, B cells, eosinophils, basophils, neutrophils, macrophages, and fibroblasts (*Dembic, 2015*; *Lacy, 2017*; *Lee et al., 2017*). We observe an increase in CCL5 levels during tumor regression and in residual tumors that is concomitant with immune cell infiltration. We therefore reasoned that the effect of CCL5 overexpression on

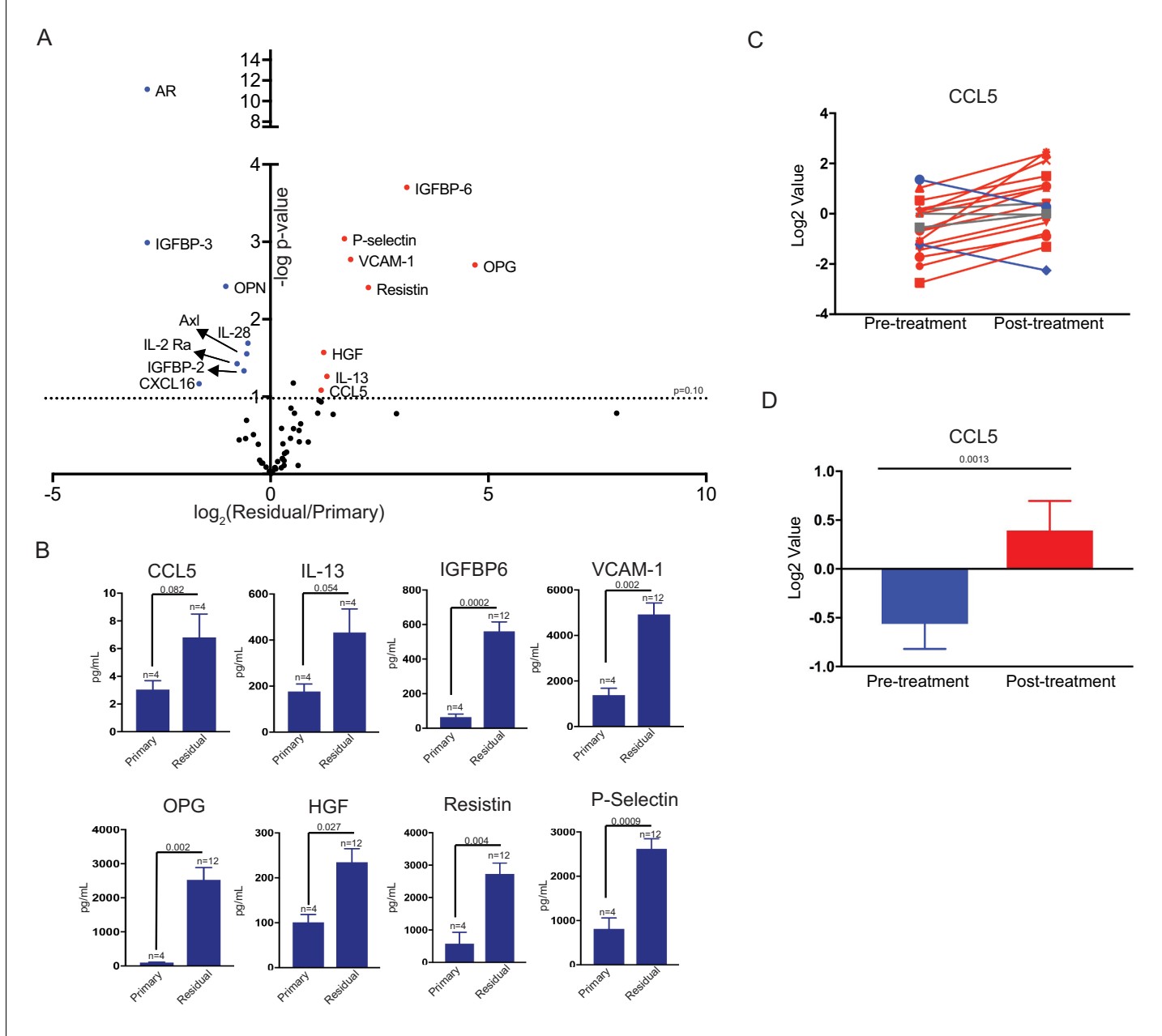

**Figure 3.** Differential cytokine expression in residual tumors. (a) Volcano plot showing differential cytokine expression between primary and residual tumors. Antibody-based cytokine arrays were used to measure cytokine expression in orthotopic primary tumors or microdissected residual tumors. Cytokines that are upregulated (fold change >2, p-value < 0.1) in dormant tumors are in red, and downregulated cytokines (fold change <-2, p-value < 0.1) are in blue. Significance was determined using a two-tailed Student's t-test. (b) Quantification of CCL5, IL-13, IGFBP6, VCAM-1, OPG, HGF, Resistin, and P-Selectin expression in primary tumors and residual tumors. Values were derived from the cytokine arrays shown in (a). Significance was determined using a two-tailed Student's t-test. (c) CCL5 expression in 18 matched pre- and post-treatment samples from GSE10281. Red lines show tumors in which CCL5 expression increased following treatment (>1.5 fold change), and blue lines show tumors with decreased CCL5 expression (<1.5 fold change). (d) Average CCL5 expression in pre- and post-treatment samples from (e). Significance was determined using a two-tailed paired Student's t-test. Error bars denote mean ± SEM.

DOI: https://doi.org/10.7554/eLife.43653.007

The following source data and figure supplement are available for figure 3:

**Source data 1.** Cytokine array expression data analysis from arrays Q1 and Q4.

DOI: https://doi.org/10.7554/eLife.43653.009

**Figure supplement 1.** Cytokine gene expression in human breast cancers following neoadjuvant therapy.

*Figure 3 continued on next page*

*Figure 3 continued*

DOI: https://doi.org/10.7554/eLife.43653.008

recurrence may be mediated through its ability to recruit one or more of these cell types to residual lesions and recurrent tumors. CCL5 can signal through multiple receptors, including CCR1, CCR3, and CCR5, but it predominately acts through CCR5 (*Soria and Ben-Baruch, 2008*). We therefore examined CCR5 expression on various immune and stromal cells in primary tumors (+dox), regressing tumors (5 days –dox), residual tumors (69 days –dox), and recurrent tumors by flow cytometry.

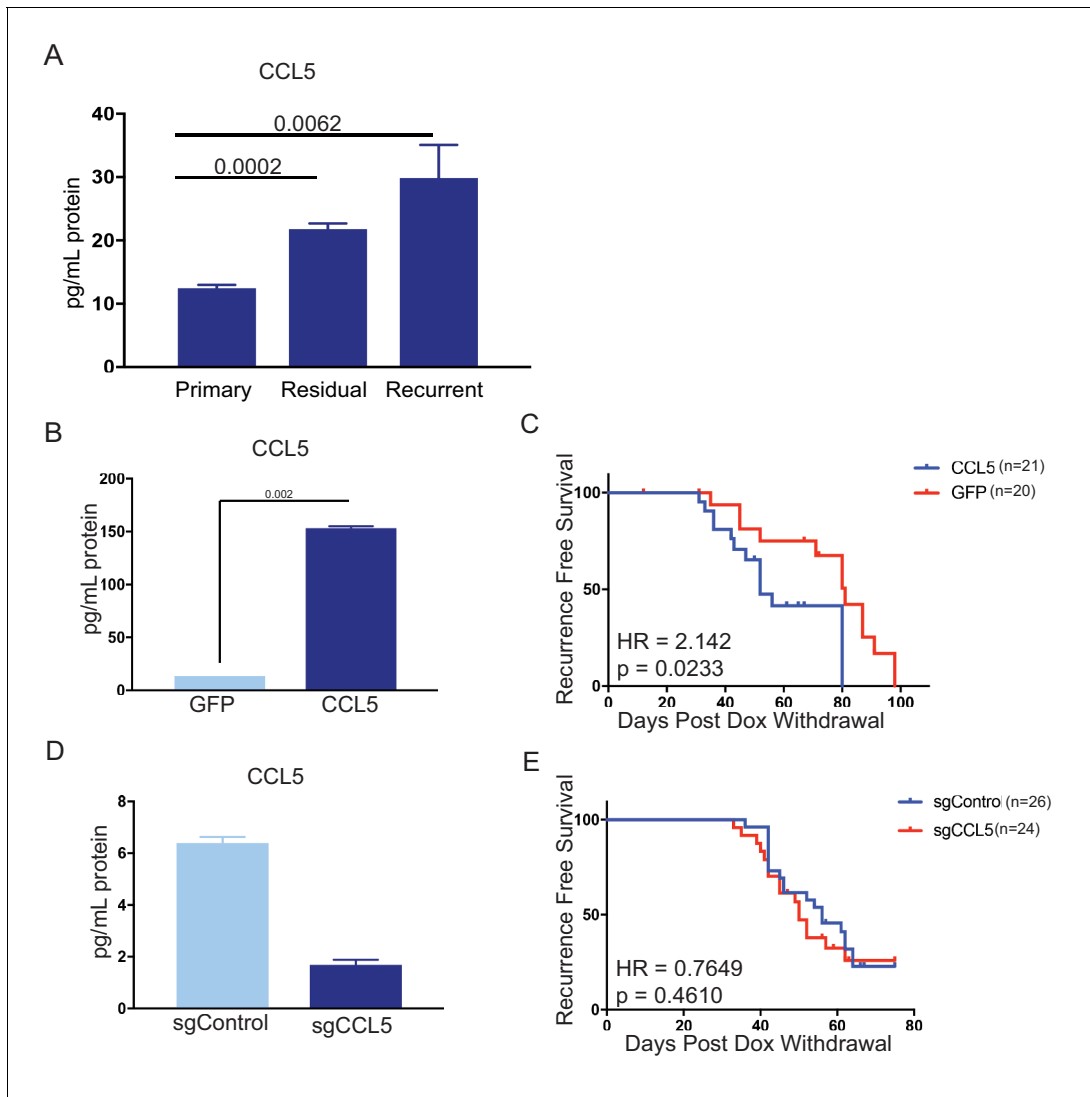

**Figure 4.** CCL5 expression promotes tumor recurrence following Her2 downregulation. (**a**) CCL5 protein levels in orthotopic primary (n = 4), residual (n = 3), and recurrent (n = 2) tumors as determined by ELISA. (**b**) CCL5 protein levels in primary tumor cells engineered to express CCL5. Results show the mean ± SEM for two independent experiments. Significance was determined using a two-tailed Student's t-test. (**c**) Recurrence-free survival for mice with control tumors or tumors expressing CCL5. CCL5 expression significantly accelerated recurrence (Hazards Ratio (HR) = 2.1, p=0.02). Results are from a single experiment with 20 control tumors and 21 CCL5 tumors. p-Values and hazards ratios are indicated. Statistical significance was determined by Mantel-Cox log rank test. (**d**) CCL5 expression as determined by ELISA in primary tumor cells expressing a control sgRNA or a sgRNA targeting CCL5. Results show the mean ± SEM for a single representative experiment. (**e**) Recurrence-free survival of mice with control tumors or CCL5 knockout tumors. CCL5 knockout in tumor cells did not significantly delay tumor recurrence (HR = 0.76, p = 0.46). Results are from a single experiment with 26 control tumors (sgControl) and 24 sgCCL5 tumors. Statistical significance was determined by Mantel-Cox log rank test. Error bars denote mean ± SEM.
DOI: https://doi.org/10.7554/eLife.43653.010

As expected, Her2 was downregulated following dox withdrawal in all tumors (*Figure 5—figure supplement 1A*). For each cell type, we measured the median fluorescence intensity (MFI) of CCR5 staining in CCR5+ cells. Interestingly, the level of CCR5 expressed on macrophages increased in residual tumors (*Figure 5A* and *Figure 5—figure supplement 2*). In contrast, CCR5 expression on CD4+ T cells CD8+ T cells increased in regressing tumors, but returned to baseline in residual

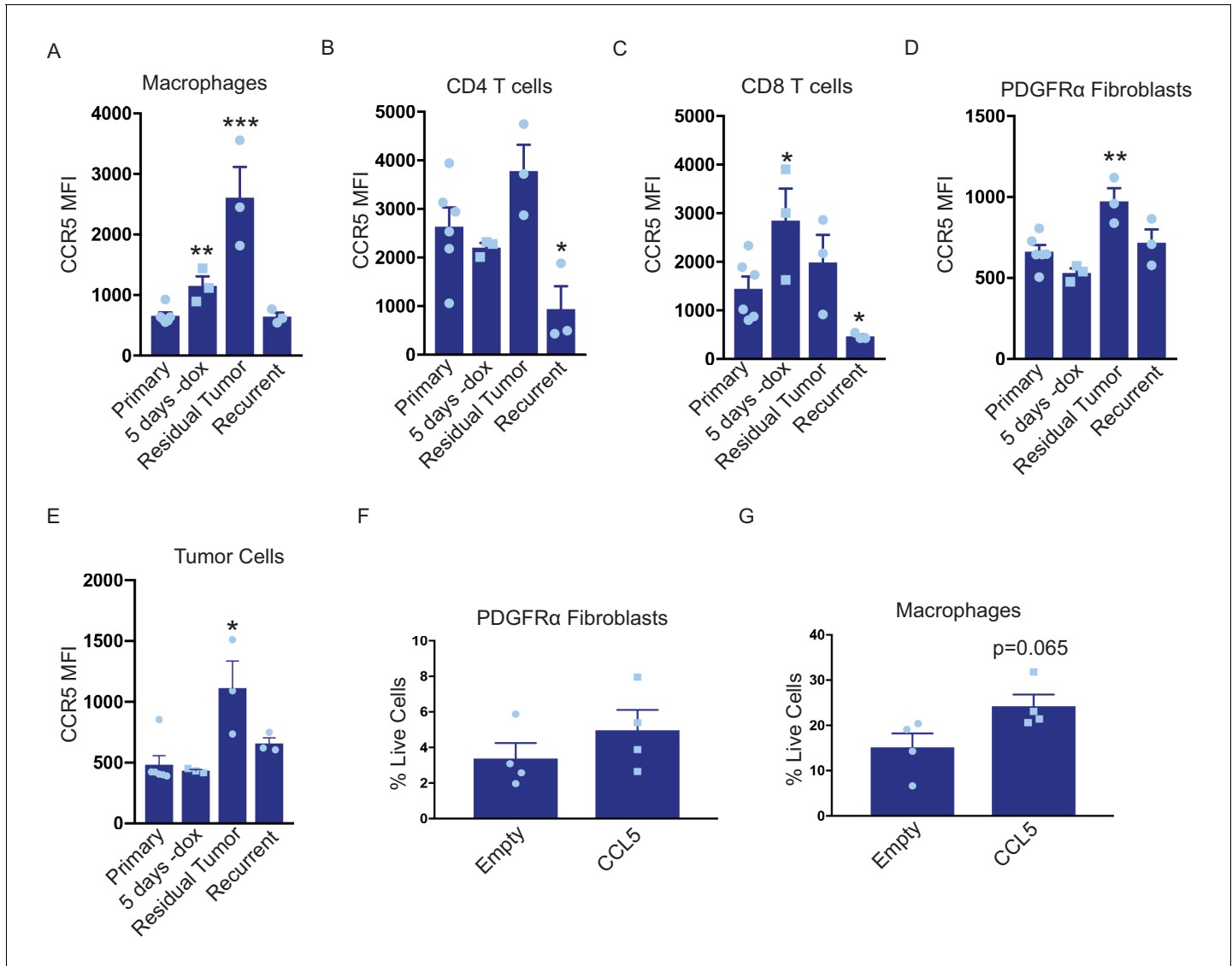

**Figure 5.** CCL5 promotes macrophage infiltration in residual tumors. (a–d) Flow cytometry of immune cells in primary (n = 6), regressing (5 days -dox; n = 3), residual (n = 3), and recurrent (n = 3) tumors from autochthonous MTB;TAN mice. Immune cell populations analyzed include CD11b+/F4/80 + macrophages (a), CD4+ T cells (b), CD8+ T cells (c), PDGFRα fibroblasts (d), and tumor cells (e). Each immune cell population was divided into CCR5- or CCR5+ cells, and the median fluorescence intensity (MFI) of the CCR5+ population was calculated. (f) Flow cytometry of CD45-/PDGFRα+ fibroblasts in control residual tumors (n = 4) or residual tumors expressing CCL5 (n = 4). (g) Flow cytometry of CD11b+/F4/80+ macrophages in control residual tumors (n = 4) or residual tumors expressing CCL5 (n = 4). Error bars denote mean ± SEM. Significance was determined using a two-tailed Student's t-test. *p < 0.05, **p < 0.01, ***p < 0.001, ****p < 0.0001.

DOI: https://doi.org/10.7554/eLife.43653.011

The following figure supplements are available for figure 5:

**Figure supplement 1.** CCL5 recruits CCR5+ macrophages to residual tumors.
DOI: https://doi.org/10.7554/eLife.43653.012
**Figure supplement 2.** CCR5 staining on immune cell populations in primary, regressing, residual, and recurrent tumors.
DOI: https://doi.org/10.7554/eLife.43653.013

tumors (*Figure 5B and C*, *Figure 5—figure supplement 2*). Similar to macrophages, the expression of CCR5 on fibroblasts was elevated in residual tumors (*Figure 5D*, *Figure 5—figure supplement 2*). We were also interested in examining CCR5 expression on CD45– tumor cells. We observed a slight increase in CCR5 expression in residual tumor cells, but otherwise there was no change in CCR5 expression on these cells (*Figure 5E*). To directly compare the expression of CCR5 in macrophages and tumor cells, we sorted these two populations from primary, regressing, residual, and recurrent tumors from MTB;TAN mice and performed qPCR analysis. CCR5 was expressed at higher levels on macrophages than tumor cells at each stage, and its expression was especially high on residual tumor macrophages (*Figure 5—figure supplement 1B*). Overall, these results identify several cell types – notably macrophages and fibroblasts – that express high levels of CCR5 and so are poised to respond to CCL5 in residual tumors.

To determine whether these cell types are recruited by CCL5 in residual tumors, we generated primary and residual tumors overexpressing CCL5 and analyzed the abundance of macrophages and fibroblasts by flow cytometry. Fibroblast levels were not significantly different between control and CCL5-expressing tumors (*Figure 5F*, *Figure 5—figure supplement 1C*). In contrast, CCL5-expressing tumors exhibited a modest but consistent increase in macrophage infiltration (*Figure 5G*, *Figure 5—figure supplement 1D*). Taken together, these results suggest that CCL5 expression in residual tumors can recruit CCR5-positive macrophages, and suggest that CCL5 may subsequently signal through CCR5 on these cells to modulate macrophage function.

## Macrophages express and secrete collagen and collagen deposition factors

We next considered the possibility that CCL5 recruitment of macrophages to residual tumors may promote recurrence through macrophage-tumor cell crosstalk. To address this, we sorted CD45+/CD11b+/F4/80+ macrophages from primary, residual and recurrent tumors from the autochthonous MTB;TAN model by fluorescence activated cell sorting (FACS), and then isolated RNA from the sorted cell populations for RNAseq. Residual tumor-associated macrophages did not yield sufficient RNA for RNAseq, but we were able to sequence RNA from primary, regressing, and recurrent tumor-associated macrophages (TAMs). Examination of differentially expressed genes between primary and recurrent TAMs suggested that FACS-sorted TAMs may have been partially contaminated with tumor cells. For instance, we detected Her2 expression at high levels in primary TAMs and low levels in recurrent TAMs. Therefore, we used a gene expression dataset of primary and recurrent tumor cells cultured in vitro to filter the TAM expression list (*Figure 6—source data 1*). After filtering, we were left with approximately 200 genes that were differentially expressed between primary and recurrent tumor macrophages (*Figure 6A*, *Figure 6—source data 2*). Interestingly, genes encoding fibrillar collagen and collagen deposition proteins were more highly expressed in the recurrent TAMs than the primary TAMs or regressing tumor TAMs (*Figure 6B*). These genes include Collagen alpha-1(V) chain (COL5A1), Collagen type XXIV alpha 1 (COL24A1), Procollagen C-endopeptidase enhancer 1 (PCOLCE), and Asporin (ASPN). COL5A1 and COL24A1 encode fibrillar collagens, PCOLCE encodes a glycoprotein that binds and drives the cleavage of type one fibrillar procollagen, and ASPN encodes a protein that binds to fibrillar collagens to regulate mineralization. We next sought to validate these findings by performing qPCR analysis on primary, regressing, residual, and recurrent TAMs. This analysis showed that the expression of these genes progressively increased during tumor regression, residual disease, and recurrence (*Figure 6C*). Additionally, qPCR on RNA isolated from bulk tumors showed higher expression of COL5A1 and COL24A1 in recurrent tumors, while a subset of recurrent tumors had high expression of ASPN and PCOLCE (*Figure 6D*). Consistent with this, Masson's trichrome staining showed increased collagen deposition in residual and recurrent tumors (*Figure 6E*, middle and bottom). In order to see if similar gene expression patterns are observed in residual disease in breast cancer patients, we examined gene expression data from residual tumors after neoadjuvant targeted therapy. Indeed, expression of these four collagen genes increased in residual tumors following therapy (*Figure 6—figure supplement 1A*). Finally, we asked whether CCL5 regulates collagen deposition by comparing collagen levels in control and CCL5-expressing recurrent tumors. While control recurrent tumors had uniform levels of collagen deposition (*Figure 6F* and *Figure 6—figure supplement 1B–C*), a subset of CCL5-expressing tumors had very high levels of collagen deposition (*Figure 6F* and *Figure 6—figure supplement 1B–C*). Taken together, these results suggest that CCL5 promotes macrophage infiltration and

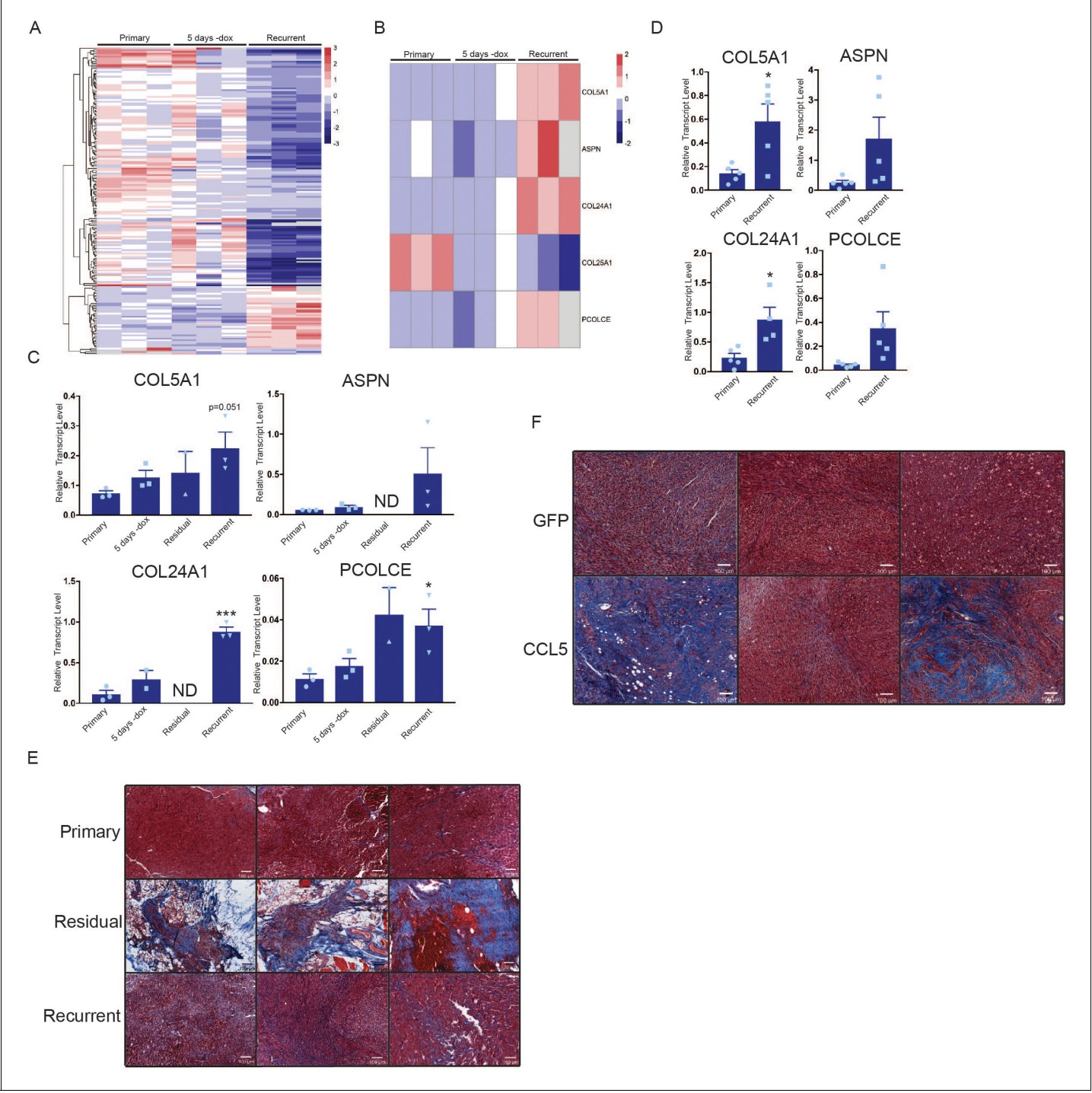

**Figure 6.** Macrophages express collagen and collagen deposition factors. (**a**) RNA-seq analysis of tumor-associated macrophages from primary (n = 3), regressing (5 days -dox; n = 3), and recurrent (n = 3) tumors. The heatmap shows differentially expressed genes (p<0.01, Student's t-test) between primary and recurrent TAMs. (**b**) Heatmap showing expression of specific collagen genes from RNA-seq analysis in (a). (**c**) qRT-PCR analysis of COL5A1, ASPN, COL24A1, and PCOLCE expression in the cohort in (a) along with sorted macrophages from residual tumors. ND = not detected (**d**) qRT-PCR analysis of COL5A1, ASPN, COL24A1, and PCOLCE expression in unsorted MTB;TAN primary (n = 5) and recurrent (n = 5) tumors. (**e**) Masson's trichrome staining showing collagen deposition in primary (n = 3), residual (n = 3), and recurrent (n = 3) tumors from the MTB;TAN model. Collagen is stained in blue, and higher collagen staining is present in residual and recurrent tumors. (**f**) Masson's trichrome staining in a subset of control and CCL5-expressing orthotopic recurrent tumors. The entire cohort of tumors is shown in *Figure 6—figure supplement 1*. Error bars denote mean ± SEM. Significance was determined using a two-tailed Student's t-test. *p < 0.05, ***p < 0.001.

DOI: https://doi.org/10.7554/eLife.43653.014

*Figure 6 continued on next page*

*Figure 6 continued*

The following source data and figure supplement are available for figure 6:

**Source data 1.** Differentially expressed genes from RNA-seq from primary and recurrent tumor cell lines used to clear contaminates from TAM RNA-seq.
DOI: https://doi.org/10.7554/eLife.43653.016
**Source data 2.** Candidate list of differnetially expressed genes between primary and recurrent TAMs after filtering.
DOI: https://doi.org/10.7554/eLife.43653.017
**Figure supplement 1.** Collagen gene expression and deposition in residual and recurrent tumors.
DOI: https://doi.org/10.7554/eLife.43653.015

collagen deposition. Given the importance of collagen for regulating tumor cell function, this may be one mechanism by which CCL5 expression accelerates recurrence. This is reminiscent of findings in colorectal cancer, where collagen deposition can be mediated in part through CCR2+ macroophages, and depletion of these macrophages inhibits tumor growth (*Afik et al., 2016*).

## Discussion

The long-term survival of residual tumor cells following therapy is a major obstacle to obtaining cures in breast cancer. Understanding the pathways that promote residual cell survival – and that induce the reactivation of these cells to generate recurrent tumors – is critical for designing therapies to prevent breast cancer relapse. There has been extensive focus on tumor cell-intrinsic pathways that allow cells to survive therapy (*Holohan et al., 2013*). However, the role of tumor cell-extrinsic factors, including the tumor microenvironment, in regulating the survival and recurrence of residual cells has not been extensively explored.

Here, we used a conditional mouse model to investigate how interactions between tumor cells and the tumor microenvironment change during tumor regression, residual disease, and recurrence, and in turn how the microenvironment regulates tumor recurrence. We found that Her2 downregulation led to induction of a pro-inflammatory gene expression program comprising a number of chemokines and cytokines, including CCL5. This program was mediated by autocrine TNFα and dependent upon IKK/NFκB signaling. Notably, a recent study identified a similar gene expression program in EGFR-mutant lung cancer following treatment with EGFR inhibitors (*Gong et al., 2018*). Consistent with this pro-inflammatory gene expression program, we observed differences in immune and stromal cell infiltration during tumor regression. Both adaptive (CD4+ and CD8+ T cells) and innate (macrophages) immune cells were recruited to regressing tumors. The residual tumor microenvironment is markedly different from that of primary tumors, with high numbers of macrophages and fibroblasts, abundant collagen deposition, and differential expression of a suite of cytokines, including CCL5. Functionally, CCL5 overexpression promotes macrophage recruitment, collagen deposition, and promotes tumor recurrence. These results identify CCL5 as a critical regulator of crosstalk between residual tumor cells and the residual tumor microenvironment that promotes tumor recurrence.

A number of studies have found that Her2 signaling directly activates the NFκB pathway, and that this is functionally important for tumor growth (*Liu et al., 2009*). Consistent with this, we observed basal levels of p65 phosphorylation in primary tumor cells. Surprisingly, we found that Her2 inhibition further activates the NFκB pathway, and that this occurs through an autocrine pathway that is likely mediated by increased TNFα expression. Hyperactivation of the NFκB pathway in turn leads to the production of a number of cytokines and chemokines which may contribute to the recruitment of immune cells. These findings are consistent with prior work showing that the NFκB pathway is required for macrophage recruitment in a similar Her2-driven mouse model (*Liu et al., 2010*). Our findings add to these previous studies by showing that Her2 inhibition leads to hyperactivation of the NFκB pathway and increased macrophage recruitment.

CCL5 has been shown to play an important role in many facets of tumor progression, such as invasion, metastasis, neoangiogenesis, and immune cell infiltration (*Aldinucci and Colombatti, 2014*). In glioblastoma, CCL5 upregulation has been correlated with recurrence in post-treatment tumors (*Hudson et al., 2018*). In triple-negative breast cancer, CCL5 expression has also been correlated with residual tumor size and tumor infiltrating lymphocytes after neoadjuvant chemotherapy (*Araujo et al., 2018*). However, CCL5 has not previously been implicated in residual cell survival or

recurrence in Her2+ or hormone receptor positive breast cancer. By analyzing gene expression datasets from breast cancer patients treated with neoadjuvant targeted or chemotherapy (*Creighton et al., 2009*; *Stickeler et al., 2011*), we show here that CCL5 expression is elevated in residual tumor cells that survive therapy. A notable observation in our study is that while CCL5 expression promoted recurrence (*Figure 4C*), knockout of CCL5 in tumor cells did not delay recurrence (*Figure 4E*). This suggests that CCL5 may be at least partially redundant with other chemokines, such as CCL2 and CXCL1 and 2, in recruiting macrophages to promote recurrence.

Mechanistically, we show that CCL5 acts to recruit CCR5+ macrophages to residual tumors, consistent with its known role as a chemoattractant factor for macrophages (*Mantovani et al., 2017*). RNAseq analysis of primary and recurrent TAMs suggested that recurrent TAMs have high expression of genes encoding fibrillar collagen and proteins required for collagen deposition. qPCR analysis indicated that residual TAMs shared this gene expression program. Consistent with this, collagen deposition is high in residual and recurrent tumors, and CCL5 expression promotes collagen deposition. Collagen deposition is traditionally thought to be driven by fibroblasts in the microenvironment (*Thannickal, 2012*). However, a recent report showed that macrophages are responsible for collagen deposition in a mouse model of colorectal cancer (*Afik et al., 2016*). Collagen deposition is important for tumor progression and invasiveness (*Provenzano et al., 2008*). Collagen bundles can potentiate cell migration and increase tissue stiffness, and enzymes which crosslink collagens are often upregulated in breast cancer and are correlated with a poor prognosis (*Lu et al., 2012*). It is possible that collagen deposition may promote the survival or proliferation of residual tumor cells, and that this mediates the effect of CCL5 on tumor recurrence.

The findings reported here suggest that efforts to block CCL5-driven macrophage infiltration and subsequent collagen deposition may have therapeutic benefit. Possible therapies include the use of Maraviroc, a CCR5 antagonist (*Velasco-Velázquez et al., 2012*), and agents that block macrophage infiltration or function, such as the CSF-1R inhibitor PLX3397 (*DeNardo et al., 2011*; *Strachan et al., 2013*; *Zhu et al., 2014*). It is also possible that, because CCL5 is sufficient but not necessary for tumor recurrence, it would be preferable to block the induction of the pro-inflammatory program that is induced following Her2 downregulation using agents targeting TNFα or the NFκB pathway.

It is important to note that while our studies focus on the function of CCL5 in recruiting CCR5 + macrophages, breast cancer cells themselves can also express CCR5. Indeed, previous studies have found that CCR5 acts in tumor cells to promote stem cell expansion and metastasis in breast cancer (*Jiao et al., 2018*; *Velasco-Velázquez et al., 2012*). Although in the current study we find that in residual tumors CCR5 is expressed at higher levels in macrophages than on tumor cells, it is possible that tumor cell-expressed CCR5 may mediate at least some of the effects of CCL5 on tumor recurrence. Future work with mice lacking CCR5 on specific cell types will clarify the relative important of CCR5 on macrophages and tumor cells.

The survival and recurrence of residual tumor cells is a critical clinical problem in breast cancer. The results identified here show that interactions between residual tumor cells and their microenvironment are critical for recurrent tumor formation. Targeting tumor cell-microenvironment interactions may hold promise for preventing recurrent breast cancer.

# Materials and methods

**Key resources table**

| Reagent type (species) or resource | Designation | Souce or reference | Identifiers | Additional information |
|---|---|---|---|---|
| Recombinant DNA reagent | pLenti CMV GFP Neo | Addgene | Plasmid # 17447 RRID:Addgene_17447 | *Campeau et al., 2009* |
| Recombinant DNA reagent | lentiCas9-Blast | Addgene | Plasmid # 52962 RRID:Addgene_52962 | *Sanjana et al., 2014* |

*Continued on next page*

*Continued*

| Reagent type (species) or resource | Designation | Souce or reference | Identifiers | Additional information |
|---|---|---|---|---|
| Recombinant DNA reagent | lentiGuide-Puro | Addgene | Plasmid # 52963 RRID:Addgene_52963 | *Sanjana et al., 2014* |
| Recombinant DNA reagent | psPAX2 | Addgene | Plasmid # 12260 RRID:Addgene_12260 | Trono Lab Packing and Envelope Plasmids |
| Recombinant DNA reagent | pMD2.G | Addgene | Plasmid# 12259 RRID:Addgene_12259 | Trono Lab Packing and Envelope Plasmids |
| Cell line (*M. musculus*) | NIH-3T3 | American Type Culture Collection | Cat# CRL-1658 RRID:CVCL_0594 | |
| Cell line (*M. musculus*) | 54074 | This paper | | Derived from MTB;TAN model |
| Cell line (*M. musculus*) | 99142 | This paper | | Derived from MTB;TAN model |
| Cell line (*H. sapiens*) | 293T Ampho | American Type Culture Collection | Cat# CRL-3213 RRID:CVCL_H716 | |
| Cell line (*H. sapiens*) | 293T Eco | American Type Culture Collection | Cat# CRL-3214 RRID:CVCL_H717 | |
| Antibody | Rabbit monoclonal anti-NFκB p65 | Cell Signaling | D14E12 RRID:AB_10859369 | 1:1000 (WB) |
| Antibody | Rabbit monoclonal anti-p-NFκB p65 | Cell Signaling | 93H1 RRID:AB_10827881 | 1:1000 (WB) |
| Antibody | Mouse monoclonal anti-Tubulin | Santa Cruz | TU-02 RRID:AB_628408 | 1:1000 (WB) |
| Antibody | Goat anti-rabbit HRP | Cell Signaling | Cat# 7074 RRID:AB_2099233 | 1:5000 (WB) |
| Antibody | Goat anti-mouse HRP | Cell Signaling | Cat# 7076 RRID:AB_330924 | 1:5000 (WB) |
| Antibody | Goat anti-rabbit Alexa Flour 680 | Life Technologies | Cat# A21076 RRID:AB_141386 | 1:5000 (WB) |
| Antibody | IRDYE 800CW Goat anti-mouse | LI-COR | Cat# 926–32210 RRID:AB_621842 | 1:5000 (WB) |
| Antibody | Rat monoclonal anti-CD45R/B220, APC conjugated | Invitrogen/ eBioscience (Carlsbad, CA) | RA3-6B2 RRID:AB_469395 | 1:50 (FC) |
| Antibody | Hamster monoclonal anti-CD49b, AF488 conjugated | BioLegend | HMα2 RRID:AB_492851 | 1:200 (FC) |
| Antibody | Hamster monoclonal anti-FcεRIα, PE conjugated | BioLegend | 1-Mar RRID:AB_1626104 | 1:50 (FC) |

*Continued on next page*

*Continued*

| Reagent type (species) or resource | Designation | Souce or reference | Identifiers | Additional information |
|---|---|---|---|---|
| Antibody | Rat monoclonal anti-Siglec-F/CD170, PE conjugated | BD | E50-2440 RRID:AB_10896143 | 1:200 (FC) |
| Antibody | Rat monoclonal anti-PDGFRα/CD140a, PE conjugated | Invitrogen/ eBioscience | APA5 RRID:AB_657615 | 1:100 (FC) |
| Antibody | Rat monoclonal anti-CD45, PECy5 conjugated | BD | 30-F11 RRID:AB_394612 | 1:200 (FC) |
| Antibody | Mouse monoclonal anti-CD45, APC conjugated | BD | 30-F11 RRID:AB_1645215 | 1:200 (FC) |
| Antibody | Rat anti-CD45, V50 conjugated | BD | 30-F11 RRID:AB_1645275 | 1:200 (FC) |
| Antibody | Rat monoclonal anti-F4/80, AF647 conjugated | BD | T45-2342 RRID:AB_2744474 | 1:50 (FC) |
| Antibody | Rat monoclonal anti-CD11b, PE conjugated | BD | M1/70 RRID:AB_394775 | 1:50 (FC) |
| Antibody | Rat monoclonal anti-CD11b, PECy7 conjugated | BD | M1/70 RRID:AB_2033994 | 1:100 (FC) |
| Antibody | Rat monoclonal anti-Ly6G, APC conjugated | BD | 1A8 RRID:AB_1727560 | 1:200 (FC) |
| Antibody | Hamster monoclonal anti-CD3e, PE conjugated | BD | 145–2 C11 RRID:AB_394460 | 1:100 (FC) |
| Antibody | Rat monoclonal anti-CD4, APCC7y conjugated | BD | GK1.5 RRID:AB_394331 | 1:100 (FC) |
| Antibody | Rat monoclonal anti-CD8a, APC conjugated | BD | 53–6.7 RRID:AB_398527 | 1:200 (FC) |
| Antibody | Rat monoclonal anti-CD16/ CD32 Fc Blocker | BD | 2.4G2 RRID:AB_394659 | 1:50 (FC) |
| Antibody | Rat monoclonal anti-CCR5/CD195, BV421 conjugated | BD | C34-3448 RRID:AB_2741677 | 1:100 (FC) |

*Continued on next page*

*Continued*

| Reagent type (species) or resource | Designation | Souce or reference | Identifiers | Additional information |
|---|---|---|---|---|
| Antibody | Mouse monoclonal anti-Cytokertin 8 | Troma 1, Brulet, P, Kemler, R Institut Pasteur, Paris, France | Troma 1 RRID:AB_ 531826 | 1:50 (IHC) |
| Antibody | Rat monoclonal anti-CD45 | BD Biosciences | 30-F11 RRID:AB_ 394606 | 1:200 (IHC) |
| Antibody | Rabbit monoclonal anti-CD3 | Themo | SP7 RRID:AB_ 1956722 | 1:100 (IHC) |
| Antibody | Rat monoclonal anti-F4/80 | Bio-Rad | Cl:A3-1 RRID:AB_ 1102558 | 1:1000 (IHC) |
| Peptide, recombinant protein | TNFα, mouse | BioLegend | Cat# 575202 | 10 ng/mL |
| Commercial assay or kit | Trichrome stain | Abcam | ab150686 | |
| Commercial assay or kit | Vectastain ABC Kit (Rabbit IgG) | Vector Labs | Cat# PK-6101 | |
| Commercial assay or kit | Vectastain ABC Kit (Rat IgG) | Vector Labs | Cat# PK-4004 | |
| Commercial assay or kit | RNeasy Mini Kit | Qiagen | Qiagen:74106 | |
| Commercial assay or kit | QIAshredder | Qiagen | Qiagen:79656 | |
| Commerical assay or kit | Quantibody Mouse Cytokine Array Q1 | RayBiotech | Cat# QAM-CYT-1–1 | |
| Commercial assay or kit | Quantibody Mouse Cytokine Array Q4 | RayBiotech | Cat# QAM-CYT-4 | |
| Chemical compound, drug | IKK16 | Selleckchem | Cat# S2882 | 100 nM |
| Chemical compound, drug | Lipofectamine 2000 | Life Technologies | Cat# 11668019 | 60 μL per reaction |
| Chemical compound, drug | Polybrene | Sigma | Cat# 107689 | 6 μg/mL |
| Chemical compound, drug | 2x Cell Lysis Buffer | RayBiotech | Cat# AA-LYS | |
| Chemical compound, drug | Luminata Classico/ Crescendo Western HRP Substrate | Millipore | Cat#WBLUC0500 Cat# WBLUR0500 | |
| Chemical compound, drug | Doxycycline | RPI | Cat# D43020-100.0 | 2 mg/kg in vivo and 2 μg/mL in vitro |

*Continued on next page*

*Continued*

| Reagent type (species) or resource | Designation | Souce or reference | Identifiers | Additional information |
|---|---|---|---|---|
| Sequence-based reagent | RT-PCR primers | This paper | CCL5 cDNA into pK1 plasmid | Forward: TAACCTCGAGATGAAGATCTCTGCAGCTG, Reverse: TAACGCGGCCGCCAGGGTCAGAATCAAGAAACC |
| Sequence-based reagent | RT-PCR primers | This paper | CCL5 cDNA into pLenti CMV plasmid | Forward: TAACTCTAGAATGAAGATCTCTGCAGCTG, Reverse: TAACGTCGACCAGGGTCAGAATCAAGAAACC |
| Sequence-based reagent | gRNAs | This paper | Targeting CCL5 | CCL5_1 (TGTAGAAATACTCCTTGACG), CCL5_2 (TACTCCTTGACGTGGGCACG), CCL5_3 (TGCAGAGGGCGGCTGCAGTG) |
| Sequence-based reagent | CCL5 | Thermo | Mm01302427_m1 | |
| Sequence-based reagent | CXCL1 | Thermo | Mm04207460_m1 | |
| Sequence-based reagent | CXCL2 | Thermo | Mm00436450_m1 | |
| Sequence-based reagent | CXCL5 | Thermo | Mm00436451_g1 | |
| Sequence-based reagent | CCL2 | Thermo | Mm00441242_m1 | |
| Sequence-based reagent | Actin | Thermo | Mm02619580_g1 | |
| Sequence-based reagent | ASPN | Thermo | Mm00445945_m1 | |
| Sequence-based reagent | PCOLCE | Thermo | Mm00476608_m1 | |
| Sequence-based reagent | COL5A1 | Thermo | Mm00489299_m1 | |
| Sequence-based reagent | COL24A1 | Thermo | Mm01323744_m1 | |
| Software, algorithm | GraphPad Prism | GraphPad Prism (https://graphpad.com) | RRID:SCR_002798 | Version 8 |
| Software, algorithm | JMP Pro | SAS Institute Inc, Cary, NC | | |
| Software, algorithm | FlowJo | TreeStar | RRID:SCR_008520 | |
| Software, algorithm | Fiji | Fiji (http://fiji.nih.gov/ | RRID:SCR_002285 | *Schindelin et al., 2012* |

WB = Western blot, FC = flow cytometry, IHC = immunohistochemistry

## Orthotopic recurrence assays

Orthotopic tumor recurrence assays were performed as described (Alvarez et al., 2013). Briefly, cohorts of 6-week-old recipient mice (nu/nu or TAN) on doxycycline were injected bilaterally in the #4 inguinal mammary fat pad with $1 \times 10^6$ primary tumor cells (expressing either a control sgRNA, a sgRNA targeting CCL5, CCL5 cDNA, or GFP cDNA). Once tumors reached 5 mm (2–3 weeks), doxycycline was removed to initiate oncogene down-regulation and tumor regression. Mice were palpated biweekly to monitor tumor recurrence, and sacrificed when recurrent tumors reached 10 mm. Differences in recurrence-free survival between control and experimental cohorts were compared using Kaplan-Meier survival curves (Kaplan and Meier, 1958) and evaluated by the p-value from a log-rank test and the hazard ratio from the Cox proportional hazard regression, as described previously (Alvarez et al., 2013).

Power calculations were used to determine cohort size for each in vivo experiment. Briefly, in order to detect a 2.5-fold difference in recurrence-free survival between control and experimental groups, given a median recurrence-free survival of 60 days for the control group and a 300 day follow-up, we estimated we would need to enroll 22 tumors per group (80% power, p<0.05). We enrolled extra mice in each cohort to account for tumor take rates and unexpected mortality. Final cohort sizes were: GFP tumors, 17 mice (34 tumors); CCL5 tumors, 18 mice (36 tumors); sgControl tumors, 20 mice (40 tumors); sgCCL5 tumors, 20 mice (40 tumors).

## Tissue culture and reagents

Cell lines derived from primary MTB;TAN tumors were grown as previously described in media containing 2 µg/ml dox (Alvarez et al., 2013). For conditioned media experiments, primary tumor cell lines were plated on 10 cm plates. 24 hr later, media was changed to media without dox, and conditioned media was collected 1 or 2 days later. Media was centrifuged to remove cells, supplemented with 2 µg/ml dox, and applied to naive primary tumor cells. Cells treated with conditioned media were harvested 1 or 2 days later for qPCR or Western blot analysis. For dox withdrawal experiments, primary tumor cell lines were plated 10 cm plates. 24 hr later, media was changed to media without dox and cells were collected 1 or 2 days later for qPCR or western blot analysis. IKK16 (Selleckchem, Houston, TX) was used at 100 nM, TNFα (BioLegend, San Diego, CA) was used at 10 ng/ml.

Primary cells derived from MTB;TAN tumors (54074 and 99142 cells) were generated by our lab, are used at early passages, and as a result have not been authenticated. NIH3T3 cells were tested by the Duke Cell Culture Facility for mycoplasma contamination and tested negative. The facility was not able to perform STR authentication on these mouse cells.

## Flow cytometry

Tumors were harvested and digested as previously described (Mabe et al., 2018). Cells were aliquoted at $1 \times 10^6$ cells per 5 mL falcon tube. CD16/CD32 Fc Block antibody was added for 10 min at 4°C (2 µL/$1 \times 10^6$ cells). Tumors were then stained with antibody cocktails listed below for 30 min at 4°C, and then washed three times with FACs buffer (BD Biosciences, Billerica, MA).

| Cell type | Antibody | Fluorophore | Clone | Vendor | Dilution |
|---|---|---|---|---|---|
| B Cell | CD45R/B220 | APC | RA3-6B2 | Invitrogen/eBioscience (Carlsbad, CA) | 1:50 |
| Basophil | CD49b | AF488 | HMα2 | BioLegend | 1:200 |
| Basophil | FcεRIα | PE | MAR-1 | BioLegend | 1:50 |
| Eosinophil | Siglec-F/CD170 | PE | E50-2440 | BD | 1:200 |
| Fibroblast | PDGFRα/CD140a | PE | APA5 | Invitrogen/eBioscience | 1:100 |
| Leukocyte | CD45 | PECy5 | 30-F11 | BD | 1:200 |
| Leukocyte | CD45 | APC | 30-F11 | BD | 1:200 |
| Leukocyte | CD45 | V450 | 30-F11 | BD | 1:200 |

*Continued on next page*

*Continued*

| Cell type | Antibody | Fluorophore | Clone | Vendor | Dilution |
|---|---|---|---|---|---|
| Macrophage | F4/80 | AF647 | T45-2342 | BD | 1:50 |
| Monocyte/ Granulocyte | CD11b | PE | M1/70 | BD | 1:50 |
| Monocyte/ Granulocyte | CD11b | PECy7 | M1/70 | BD | 1:100 |
| Neutrophil | Ly6G | APC | 1A8 | BD | 1:200 |
| T Cell | CD3e | PE | 145–2 C11 | BD | 1:100 |
| T Cell | CD4 | APCCy7 | GK1.5 | BD | 1:100 |
| T Cell | CD8a | APC | 53–6.7 | BD | 1:200 |
| - | Fc Blocker | - | 2.4G2 | BD | 1:50 |
| - | CCR5/CD195 | BV421 | C34-3448 | BD | 1:100 |

Cells were analyzed using a FACSCanto analyzer (BD Biosciences) and data were analyzed using FlowJo software (TreeStar, Ashland, OR). Gating of the CCR5-high population was determined by using a fluorescence minus one (FMO; cells stained with antibodies for cell type markers, lacking the CCR5 antibody) histogram in the fluorescence channel for the CCR5 antibody as a negative control. The FMO negative control histogram was plotted with a positive control of the single stain (cells stained only with CCR5 antibody) from the same tumor. Percent of CCR5+ cells were gated according to the positive control.

## qPCR

RNA was isolated from tumors and cells using RNeasy columns (Qiagen, Hilden, Germany). 1 μg of RNA was reversed transcribed using cDNA synthesis reagents (Promega, Madison, WI). qPCR was performed using 6-carboxyfluorescein labeled TaqMan probes (Thermo, Waltham, MA): CCL5 (Mm01302427_m1), CXCL1 (Mm04207460_m1), CXCL2 (Mm00436450_m1), CXCL5 (Mm00436451_g1), CCL2 (Mm00441242_m1), Actin (Mm02619580_g1), ASPN (Mm00445945_m1), PCOLCE (Mm00476608_m1), COL5A1 (Mm00489299_m1), COL24A1 (Mm01323744_m1), and read on a Bio-Rad (Hercules, CA) CFX qPCR machine.

## Western blotting and cytokine arrays

Western blotting was performed as described (*Alvarez et al., 2013*) using the following antibodies: NFκB p65 (D14E12, Cell Signaling, Danvers, MA), p-NFκB p65 (93H1, Cell Signaling), and tubulin (TU-02, Santa Cruz, Dallas, TX), all at a 1:1000 dilution. Secondary antibodies conjugated to Alexa Flour 680 (Life Technologies, Carlsbad, CA) or 800 (LI-COR Biosciences, Lincoln, NE) were detected with the Odyssey detection system (LI-COR Biosciences). For p-p65 detection, secondary antibodies conjugated to HRP were used and blots were developed using Classico or Crescendo reagent (Millipore, Burlington, MA) and exposed to film (VWR, Radnor, PA). Secondary antibodies were used at a 1:5000 dilution.

For cytokine array analysis, tumor lysates were made in 2X lysis buffer (RayBiotech, Norcross, GA) and diluted to 50 μg per 100 μL in diluent provided. Tumor lysates and standards were run on both Quantibody Mouse Cytokine Array Q1 and Q4 (RayBiotech). Slides were scanned and quantified by RayBiotech.

## Plasmids and CRISPR/Cas9

pLenti CMV GFP Puro was purchased from Addgene (Watertown, MA).

A CCL5 cDNA encoding the full-length mouse protein was amplified by RT-PCR from recurrent MTB;TAN tumor cells and cloned into the retroviral expression vector pK1 using the following primers: Forward: TAACCTCGAGATGAAGATCTCTGCAGCTG, Reverse: TAACGCGGCCGCCAGGG TCAGAATCAAGAAACC.

A CCL5 cDNA encoding the full-length mouse protein was amplified by RT-PCR from recurrent MTB;TAN tumor cells and cloned into the lentiviral expression vector pLenti CMV using the following

Cancer Biology | Immunology and Inflammation

primers: Forward: TAACTCTAGAATGAAGATCTCTGCAGCTG, Reverse: TAACGTCGACCAGGG TCAGAATCAAGAAACC.

CCL5 CRISPR sgRNAs: CCL5_1 (TGTAGAAATACTCCTTGACG), CCL5_2 (TACTCCTTGACG TGGGCACG), CCL5_3 (TGCAGAGGGCGGCTGCAGTG). A small guide against AAVS was used as control. sgRNAs were cloned into Lentiguide puro (*Sanjana et al., 2014*). Cas9 infection was with lentiguide Cas9 blast (*Sanjana et al., 2014*).

Retrovirus was produced by transfecting the packaging lines 293T Ampho and 293T Eco with the retroviral construct pK1 empty or CCL5 using Lipofectamine 2000. Retroviral supernatant was collected 48 hr post-transfection, filtered, and used to transduce cells in the presence of 6 µg/mL polybrene (Sigma, St. Louis, MO).

Lentivirus was produced by transfecting 293 T cells with the packaging plasmids psPAX2 and pMD2.G and lentiviral construct pLenti CMV GFP or CCL5 using Lipofectamine 2000. Lentiviral supernatant was collected 48 hr post-transfection, filtered, and used to transduce cells in the presence of 6 µg/mL polybrene (Sigma).

## RNA sequencing

RNA was isolated from tumors or tumor cells using RNeasy columns (Qiagen). For TAM sequencing, macrophages were isolated by FACS using the antibody panel described above, and RNA was isolated using RNeasy columns (Qiagen). RNA was sequenced using the Illumina HiSeq 4000 libraries and sequencing platform with 50 base pair single end reads by the Duke GCB Sequencing and Genomic Technologies Shared Resource (Durham, NC). Sequencing data have been deposited in SRA as PRJNA506006 for cell line data and PRJNA505845 for macrophage data.

## Human breast cancer microarray data

Publicly available microarray data from human primary and residual breast cancer datasets GSE10281 and GSE21974 and their corresponding clinical annotation were downloaded, converted to log2 scale, and median centered. Heatmaps were created using R (*R Development Core Team, 2013*).

## Immunohistochemistry and staining

Tumor sections were fixed in 10% normal formalin for 16 hr, then washed twice with PBS and transferred to 70% ethanol for storage. Stored tumor sections were paraffin imbedded and cut on the microtome in 5 µm sections. Sections were stained using a regressive H and E protocol, immunohistochemistry, or Masson's Trichrome.

The regressive H and E protocol is as follows: dewax and rehydrate slides. Incubate slides in Harris Modified Hematoxylin with Acetic Acid (Fisher, Hampton, NH) for 5 min. Incubate in Eosin (Sigma) for 1:30 min. Then dehydrate slides and mount slides with permount and coverslip. Let dry overnight.

For cytokeratin eight staining (Troma 1, Brulet, P., Kemler, R. Institut Pasteur, Paris, France) immunohistochemistry slides were dewaxed and rehydrated as above. Slides were boiled in antigen retrieval buffer (1X in ddH$_2$O) for 5 min and allowed to cool. Slides were washed in PBS and then incubated in 0.3% H$_2$O$_2$. Slides were washed, blocked and stained according to the protocol from the rabbit secondary Vectastain ABC kit (Vector Labs, Burlingame, CA). Primary antibody was used at a dilution of 1:50. CD45 (30-F11, BD Biosciences, 1:200), CD3 (SP7, Thermo, 1:100), and F4/80 (Cl:A3-1, Bio-Rad, 1:1000) staining were performed by the Duke Pathology core (Durham, NC).

Trichrome stain was performed using a staining kit from Abcam (Cambridge, UK) (ab150686).

## Quantifying IHC and Masson's Trichrome in Fiji

To quantify the amount of positive staining for CD3, CD45, and F4/80 and for Masson's Trichrome, we used Fiji (*Schindelin et al., 2012*). The 'Color Deconvolution' function was used to separate the colors into positive staining and hematoxylin for normalization. We then converted each image to 8-bit and applied a threshold of positive staining to each image and used this same threshold across all images. We then measured the pixel area of the positive staining and normalized this to the hematoxylin staining for each image. For the primary tumors and 5 day -dox tumors, the whole

image was used for quantification. For residual tumors, we manually selected regions-of-interest to exclude adipose tissue from the quantification.

## Statistical reporting

For GSEA, the normalized enrichment score (NES) is reported. The normalized enrichment score accounts for differences in gene set size and in correlations between gene sets. The NES is based on all dataset permutations, to correct for multiple hypothesis testing. The nominal p value is also reported and is the statistical significance of the enrichment score, without adjustment for gene set size or multiple hypothesis testing. A reported p value of zero (0.0) indicates an actual p-value of less than 1/number-of-permutations. (*Subramanian et al., 2005*).

Two-tailed unpaired t-tests were used to analyze significance between primary tumor samples and all other time points for qPCR, cytokine array, and flow cytometry analysis. For the cytokine array, appropriate same size was calculated using JMP Pro (SAS Institute Inc, Cary, NC). A standard deviation of 20% was assumed, with a power of 0.8, fold change of 2, and p-value (alpha) of 0.05. This power calculation indicated that a sample size of 8 (4 tumors per cohort) was required. The same parameters were used for sample size calculation for flow cytometry analysis of control and CCL5-expressing tumors. For recurrence free survival (RFS), statistical analysis methods are listed in orthotopic recurrence assays.

Outliers were never excluded except for in flow cytometry experiments. Tumors that were >90% CD45+ were excluded from analysis to avoid analyzing tumors with potential contamination from the inguinal lymph node. For all other experiments where no power analysis was used, sample size was chosen based upon previous experience (*Alvarez et al., 2013*).

## Study approval

Animal care and all animal experiments were performed with the approval of and in accordance with Duke University IACUC guidelines. Mice were housed under barrier conditions.

# Acknowledgements

We thank Cui Rong (Duke-NUS, Singapore) for providing technical assistance, as well as members of the Alvarez lab for providing assistance and helpful discussions. We thank Dr. Mike Cook (Duke University) and Dr. Brent Hanks (Duke University) for assistance with flow cytometry. We thank Dr. So Young Kim (Duke University) for reagents for the CRISPR-Cas9 cell lines. We also thank Dr. Donald McDonnell, Dr. Binita Das, and Dr. Ching-Yi Chang (Duke University) for providing assistance and reagents for flow cytometry.

# Additional information

### Funding

| Funder | Grant reference number | Author |
| --- | --- | --- |
| National Cancer Institute | F31 CA220957 | Andrea Walens |
| National Cancer Institute | R01 CA208042 | James V Alvarez |
| Duke University School of Medicine | | James V Alvarez |

The funders had no role in study design, data collection and interpretation, or the decision to submit the work for publication.

### Author contributions

Andrea Walens, Conceptualization, Data curation, Formal analysis, Supervision, Funding acquisition, Validation, Investigation, Visualization, Methodology, Writing—original draft, Project administration; Ashley V DiMarco, Data curation, Formal analysis, Validation, Investigation, Writing—review and editing; Ryan Lupo, Resources, Writing—review and editing; Benjamin R Kroger, Data curation, Investigation; Jeffrey S Damrauer, Data curation, Writing—review and editing; James V Alvarez,

Conceptualization, Supervision, Funding acquisition, Project administration, Writing—review and editing

### Author ORCIDs
Jeffrey S Damrauer https://orcid.org/0000-0001-8148-0285
James V Alvarez https://orcid.org/0000-0003-2910-7621

### Ethics
Animal experimentation: All animal experiments were performed with approval from the Duke institutional animal care and use committee (IACUC) under Protocol #A199-17-08 and in accordance with recommendations in the Guide for the Care and Use of Laboratory Animals of the National Institutes of Health. Mice were housed under barrier conditions with standard 12-hour light/dark hours, and fed standard chow.

### Decision letter and Author response
Decision letter https://doi.org/10.7554/eLife.43653.028
Author response https://doi.org/10.7554/eLife.43653.029

## Additional files
### Supplementary files
• Transparent reporting form
DOI: https://doi.org/10.7554/eLife.43653.019

### Data availability
Sequencing data have been deposited in SRA as PRJNA506006 for cell line data and PRJNA505845 for macrophage data.

The following datasets were generated:

| Author(s) | Year | Dataset title | Dataset URL | Database and Identifier |
|---|---|---|---|---|
| Walens A | 2018 | Tumor associated macrophage sequencing from primary, regressing, and recurrent MTB;TAN tumors. | https://www.ncbi.nlm.nih.gov/bioproject/PRJNA505845/ | NCBI Sequence Read Archive, PRJNA505845 |
| Walens A, DiMarco AV, Kroger BR, Damrauer JS, Lupo R | 2018 | Changes in gene expression after Her2 down regulation | https://www.ncbi.nlm.nih.gov/bioproject/PRJNA506006/ | NCBI Sequence Read Archive, PRJNA506006 |

The following previously published datasets were used:

| Author(s) | Year | Dataset title | Dataset URL | Database and Identifier |
|---|---|---|---|---|
| Creighton CJ, Li X, Landis M, Dixon JM et al | 2009 | Letrozole (Femara) early response to treatment | https://www.ncbi.nlm.nih.gov/geo/query/acc.cgi?acc=GSE10281 | NCBI Gene Expression Omnibus, GSE10281 |
| Stickeler E, Pils D, Klar M | 2011 | Molecular Subtype Predicts Response to Neoadjuvant Chemotherapy in Breast Cancer | https://www.ncbi.nlm.nih.gov/geo/query/acc.cgi?acc=GSE21974 | NCBI Gene Expression Omnibus, GSE21974 |

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
