## [Decision Letter]

Thank you for submitting your article "CCL5 promotes breast cancer recurrence through macrophage recruitment in residual tumor " to *eLife* for consideration as a Research Article. Your article has been reviewed by three peer reviewers, including Päivi Ojala as the Reviewing Editor and Reviewer #1, and the evaluation has been overseen by Tadatsugu Taniguchi as the Senior Editor.

The reviewers have discussed the reviews with one another and the Reviewing Editor has drafted this decision to help you prepare a revised submission.

Summary:

In a conditional mouse model of Her2-driven breast cancer Walens et al. purports to show that Her2 downregulation in the cancer cells leads to an inflammatory program and they start to express a TNFa-NFkB-driven chemokine CCL5 thereby promoting immune cell infiltration in regressing and residual tumors. This increases the survival and recurrence of dormant residual breast cancer cells through the recruitment of tumor-associated macrophages (TAMs). The results are consistent with many reports that the progression of breast and other cancers involves macrophage activation and recruitment. By utilizing an orthotopic primary tumor model of GFP- or CCL5-transduced Her2-driven tumor cells, they demonstrate that CCL5 is sufficient but not necessary to shorten time to recurrence, and that the recruited TAMs express high levels of collagen thereby possibly contributing to the observed high collagen deposition in the recurrent tumors. Overall the authors need to separate out their original work from the previously published studies, by citing the prior studies by others that are directly relevant to their work. Moreover, the impact of the manuscript would be substantially improved by finding a more specific, direct mechanistic connection to Her2, and further evidence of clinical and human relevance.

Essential revisions:

1) In the experimental setup the authors use + dox, which regulates the transgene (Her2/Erbb2) expression, and draw conclusions about NFkB signaling. However, tetracyclines per se inhibit NFkB profoundly, and have been shown to recapitulate many of the effects seen by the authors upon Her2 expression in the + dox samples. Therefore, further evidence that the effect of Her2 is to inhibit NFkB is required.

2) As most of the non-Tet system effects occur at µg/ml, and most of the Tet-system specific regulation occurs at the ng/ml level, the authors should address this important control, by confirming at which levels of dox they are seeing their effects.

3) The authors first conclusion that Her2 downregulation in mammary tumors increases NFkB mediated tumor inflammation is the opposite to current thinking based on previous studies which had shown, (using ponasterone inducible IKBSra transgenic mice and cell lines derived from such mice), that Her2 in mammary tumors induces NFkB and thereby recruits F4/80+ macrophages (PMID:19349372, PMID:21159656). This principle has been supported in about 400 subsequent citations. Thus, it is important to determine the validity of each step in the new claims and it is also important for the authors to place their findings in the context of the current understanding of this field and provide a clear explanation of the reason for the differences. For example, dox is known to inhibit NFkB signaling including most of the cytokines noted in the current study (Tang et al., Molecular Cancer 2017, 16:36 and references therein; DOI: 10.1007/s12640-015-9592-2 and PMID: 27732942). In order to show that downregulation of Her2 causes an increase in NFkB, more direct evidence is needed including but not limited to:. 1) 1A. (1C,D) Her2 levels should be shown. 2) 1F + dox conditioned medium control must be shown, 3). Evidence that the inhibition of NFkB (Figure 1) is not due to the known ability of dox to inhibit NFKB signaling. 4). Reference prior original literature i.e. Her2 induces E2F signaling (original reference; Lee et al. 1999 (DOI: 10.1128/MCB.20.2.672-683.2000).

4) It is not clear why the authors initially focused their attention on CCL5 in the validation experiment described in Figure 1D as CCL5 was not among the most differentially regulated genes in the RNASeq analysis shown in Figure 1B-C. Moreover, in the cytokine profiling assays (Figure 3A) CCL5 was again expressed at the lowest levels (3-6 pg/ml) compared to the others which were 100-1000-fold more abundant. Therefore, it would be necessary to explain and justify why they chose to concentrate on CCL5, and not the other more abundant chemokines in the inflammatory signature found in residual tumors after Her2 down-regulation.

5) In relation to CCR5 in macrophages, CCR5 is known to be expressed in breast cancer epithelial cells, TAMs and CAFs. Analysis of 2,200 human breast cancers showed a substantial number of Her2+ breast cancers expressing CCL5 and CCR5 and that CCR5+ cells give rise to new tumors and metastasis when compared with CCR5- cells (PMID:29358169, PMID:22637726). In drawing their conclusions about CCR5 on macrophages being central, the authors need to determine CCR5 expression in their breast cancer cells (at the very least in Figure 2). Moreover, in Figure 2B-D the data needs to be quantified.

6) Figure 3—figure supplement 1. Shows expression of several genes in patients treated with aromatase inhibitors. The P values need to be cited, and relevance to the current studies need to be clarified.

7) Figure 5 is thought to reflect the relationship of Her2 and CCR5 in the tumor. What is CCR5 expression in the breast cancer cells herein? The assumption of the studies is that Her2 levels changes with dox withdrawal. What is Her2 abundance in primary tumor, 5 days after dox withdrawal, residual tumor and recurrence? The proportion of CCR5+ macrophages increases as does the mean intensity of fluorescence. Does MFI reflect the receptor number? 5 days after dox removal, assuming Her2 abundance is decreased, the proportion of CCR5+ and CCR5- cells' MFI is discordant for the cells examined (macrophages, CD4^+^T cells, CD8^+^ T cells and fibroblasts). Please explain and provide similar data for the CCR5 on the breast cancer cells.

8) The text describing the collagen deposition in the control (GFP) tumors as compared to the CCL5-expressing recurrent tumors in Figure 6F (subsection “Macrophages express and secrete collagen and collagen deposition factors”) is a bit controversial. Shouldn't abundant collagen deposition be seen also in the recurrent control tumors if they are due to CCL5-attracted TAMS as shown in Figures 2D and 5A? One would assume that CCL5-overexpressing tumors just have more.

9) The authors statement that "CCL5 had not previously been implicated in residual cell survival and recurrence" is not entirely accurate. E.g. Araujo et al. (Sci. Rep 2018; PMID:29559701) showed that in triple negative breast cancer tumor infiltrating lymphocyte recruitment was correlated with residual tumor size after neoadjuvant chemotherapy (NAC) and CCL5 expression. Moreover, Hudson et al. (Front. Oncol. 2018; PMID: 30151353) have recently shown a correlation between glioblastoma recurrence and CCL5 up-regulation in the post-treatment tumors. This statement should be rephrased accordingly.

---

## [Author Response]

Essential revisions:1) In the experimental setup the authors use + dox, which regulates the transgene (Her2/Erbb2) expression, and draw conclusions about NFkB signaling. However, tetracyclines per se inhibit NFkB profoundly, and have been shown to recapitulate many of the effects seen by the authors upon Her2 expression in the + dox samples. Therefore, further evidence that the effect of Her2 is to inhibit NFkB is required.

We appreciate the reviewers raising this important point, and we agree it is critical to demonstrate that the NFkB pathway activation we observe following dox withdrawal is due to loss of Her2 signaling, and not a consequence of doxycycline inhibition of the NFkB pathway. We have performed two new experiments that we believe address this point, and support the conclusion that Her2 inhibition leads to NFkB pathway activation:

i) We have used the small-molecule dual Her2/Her3 inhibitor Neratinib as an alternative means of inhibiting Her2 signaling. These results demonstrate that, even in the presence of doxycycline, Her2 inhibition leads to induction of phospho-p65 and an increase in NFkB target genes CCL5, TNFa, and CXCL5. These results are shown in Figure 1—figure supplement 1C-F.

ii) We have treated NIH 3T3 cells with TNFa in the presence or absence of 2 ug/ml doxycycline, which is the dox concentration we use to culture primary Her2-driven tumor cells in vitro. These results demonstrate that this low dox concentration has no effect on TNFa-mediated expression of NFkB target genes. These results are shown in Figure 1—figure supplement 1G.

iii) Finally, we note that the experimental design for the conditioned media experiments presented in Figure 1D and 1F suggest that NFkB pathway activation is due to Her2 inhibition rather than due to a direct effect of dox on NFkB pathway. Specifically, in these experiments we conditioned media in the absence of dox, collected this conditioned media, and then replenished the media with dox prior to adding it to primary tumor cells. This indicates that Her2 inhibition leads to the production of a secreted factor(s) that can activate the NFkB pathway even in the presence of doxycycline. We have clarified the experimental setup and elaborated upon this point on Pages 6-7. In addition, we have added qRT-PCR analysis of Her2 transgene levels in this conditioned media experiment to Figure 1—figure supplement 1H to demonstrate that dox replenishment maintains Her2 levels in target cells.

2) As most of the non-Tet system effects occur at µg/ml, and most of the Tet-system specific regulation occurs at the ng/ml level, the authors should address this important control, by confirming at which levels of dox they are seeing their effects.

We have included the dox concentrations used for these experiments (Page 6). For in vitro experiments, dox was used at 2 μg/ml. This is well below the dox concentrations that have been shown to inhibit NFkB signaling. For instance, in Santa-Cecília et al. (PMID 26745968) the lowest concentration of dox that inhibits the NFkB pathway is 200 μM, which is 103 μg/ml (see Figure 5 of Santa-Cecília et al.). In Alexander-Savino et al. (PMID 27732942) dox was used at concentrations between 5-40 μg/ml, and the most potent inhibition of NFkB targets (e.g. Bcl2, see Figure 6A in Alexander-Savino et al.) occurred at 40 μg/ml. These concentrations are 8-fold to 50-fold higher than the concentration we use to culture primary tumor cells. The lowest concentrations of dox that have been shown to inhibit NFkB pathway are from Tang et al. (PMID: 28178994). This paper showed that 5 ug/ml dox led to a ~20% decrease in basal levels of the NFkB target gene CCL2 levels, and did not affect LPA-induced CCL2 levels (see Figure 4 in Tang et al.). In addition, this concentration of dox had very modest effects on nuclear translocation of NFkB (see Figure 5 in Tang et al.). In contrast, dox withdrawal in primary Her2-driven tumor cells led to ~10-fold induction of CCL2 (Figure 1G) and a marked increase in p65 phosphorylation (Figure 1F). In combination with the results described in Point #1 above, we believe this argues that the NFkB pathway activation observed following dox withdrawal is due to loss of Her2 signaling rather than a consequence of dox-mediated inhibition of the NFkb pathway.

3) The authors first conclusion that Her2 downregulation in mammary tumors increases NFkB mediated tumor inflammation is the opposite to current thinking based on previous studies which had shown, (using ponasterone inducible IKBSra transgenic mice and cell lines derived from such mice), that Her2 in mammary tumors induces NFkB and thereby recruits F4/80+ macrophages (PMID:19349372, PMID:21159656). This principle has been supported in about 400 subsequent citations. Thus, it is important to determine the validity of each step in the new claims and it is also important for the authors to place their findings in the context of the current understanding of this field and provide a clear explanation of the reason for the differences. For example, dox is known to inhibit NFkB signaling including most of the cytokines noted in the current study (Tang et al., Molecular Cancer 2017, 16:36 and references therein; DOI: 10.1007/s12640-015-9592-2 and PMID: 27732942). In order to show that downregulation of Her2 causes an increase in NFkB, more direct evidence is needed including but not limited to:. 1) 1A. (1C, D) Her2 levels should be shown. 2) 1F + dox conditioned medium control must be shown, 3). Evidence that the inhibition of NFkB (Figure 1) is not due to the known ability of dox to inhibit NFKB signaling. 4). Reference prior original literature i.e. Her2 induces E2F signaling (original reference; Lee et al., 1999 (DOI: 10.1128/MCB.20.2.672-683.2000)

We agree with the reviewers that our finding that Her2 inhibition leads to NFkB pathway activation seems on its face to be incompatible with the literature cited above. Specifically, Liu et al., 2009 (PMID 19349372) nicely show that Her2 activation leads to an increase in NFkB transcriptional activity, and that this is required for transformation in vitro and tumor growth in vivo. However, we note that we do observe basal p65 phosphorylation in Her2-driven tumor cells (see Figure 1F and Figure 1—figure supplement 1C). Based on this, we propose a model wherein Her2-driven mammary tumor cells have NFkB pathway activation, consistent with previous literature. Her2 inhibition (through either dox withdrawal or Neratinib treatment) leads to TNFa secretion and *hyperactivation* of the NFkB pathway. We have added text to the Discussion (third paragraph) that both cites this very important literature, and attempts to put our findings in the context of what was previously known about the relationship between Her2 and NFkB.

In addition, we agree that each of the specific points raised here by the reviewers is important. We address each point individually:

Subpoint #1: We have included qRT-PCR data showing Her2 transgene expression for the experiment shown in Figure 1A, 1C, and 1D. These results are shown in Figure 1—figure supplement 1A.

Subpoint #2: We have performed the conditioned media experiment as requested. These results are shown in Figure 1—figure supplement 1I, and support our conclusion that the NFkB pathway is activated by conditioned media from cells following Her2 downregulation but from cells cultured with Her2 signaling on.

Subpoint #3: Please see our response to Point #2, above.

Subpoint #4: We apologize for the omission. We have added this reference to the Results section.

4) It is not clear why the authors initially focused their attention on CCL5 in the validation experiment described in Figure 1D as CCL5 was not among the most differentially regulated genes in the RNASeq analysis shown in Figure 1B-C. Moreover, in the cytokine profiling assays (Figure 3A) CCL5 was again expressed at the lowest levels (3-6 pg/ml) compared to the others which were 100-1000-fold more abundant. Therefore, it would be necessary to explain and justify why they chose to concentrate on CCL5, and not the other more abundant chemokines in the inflammatory signature found in residual tumors after Her2 down-regulation.

We appreciate this point, and have attempted to clarify our reasoning for focusing on CCL5. We were interested in cytokines that fulfilled two criteria: (1) were acutely upregulated following Her2 downregulation (Figure 1); and (2) were persistently elevated in residual tumors compared to primary tumors (Figure 3). We reasoned that these cytokines were most likely to influence the behavior of tumors throughout the duration of residual disease and recurrence. Only two cytokines – CCL5 and OPG – fulfilled both criteria. Given that the function of OPG in regulating dormant cell proliferation has been described, we focused on CCL5. We have explained our reasoning in the second paragraph of the subsection “Cytokine profiling of residual tumors”.

5) In relation to CCR5 in macrophages, CCR5 is known to be expressed in breast cancer epithelial cells, TAMs and CAFs. Analysis of 2,200 human breast cancers showed a substantial number of Her2+ breast cancers expressing CCL5 and CCR5 and that CCR5+ cells give rise to new tumors and metastasis when compared with CCR5- cells (PMID:29358169, PMID:22637726). In drawing their conclusions about CCR5 on macrophages being central, the authors need to determine CCR5 expression in their breast cancer cells (at the very least in Figure 2). Moreover, in Figure 2B-D the data needs to be quantified.

We appreciate the reviewers raising this important point. To address this, we directly compared CCR5 expression between macrophages (CD45+ / CD11b+ / F4/80+ cells) and tumor cells. (We have found that the vast majority (>90%) of CD45– cells are tumor cells (data not shown)). We used qRT-PCR on sorted populations of macrophages and tumor cells from primary, regressing, residual, and recurrent tumors. These data show that CCR5 is consistently expressed at higher levels in macrophages as compared to tumor cells, though tumor cells from both regressing and residual tumors do express some CCR5. These data are shown in Figure 5—figure supplement 1B. While these data support the importance of macrophage-expressed CCR5, we agree that our data do not exclude a role for CCR5 expression on tumor cells as mediating some of the effects of CCL5. We have added text to the Discussion highlighting this fact, and citing the relevant papers mentioned by the reviewers.

6) Figure 3—figure supplement 1. Shows expression of several genes in patients treated with aromatase inhibitors. The P values need to be cited, and relevance to the current studies need to be clarified.

We have added P-values to Figure 3—figure supplement 1, and expanded upon the relevance of the gene expression changes in residual human breast cancers in the last paragraph of the subsection “Cytokine profiling of residual tumors”.

7) Figure 5 is thought to reflect the relationship of Her2 and CCR5 in the tumor. What is CCR5 expression in the breast cancer cells herein? The assumption of the studies is that Her2 levels changes with dox withdrawal. What is Her2 abundance in primary tumor, 5 days after dox withdrawal, residual tumor and recurrence? The proportion of CCR5+ macrophages increases as does the mean intensity of fluorescence. Does MFI reflect the receptor number? 5 days after dox removal, assuming Her2 abundance is decreased, the proportion of CCR5+ and CCR5- cells' MFI is discordant for the cells examined (macrophages, CD4^+^T cells, CD8^+^ T cells and fibroblasts). Please explain and provide similar data for the CCR5 on the breast cancer cells.

In Figure 5, we intended to demonstrate that there is an increase in CCR5 expression on macrophages in residual tumors. We apologize for any confusion caused by the presentation of the data. To clarify these results, and to address the reviewers’ questions, we have made the following changes:

- We have added a panel showing CCR5 expression on tumor cells to Figure 5E, and included the flow histograms in Figure 5—figure supplement 2.

- We have performed qRT-PCR for the Her2 transgene on primary, regressing (5d –dox), residual, and recurrent tumors to confirm Her2 downregulation; this is shown in Figure 5—figure supplement 1A.

- As described in Point #5 above, we also measured CCR5 expression on tumor cells using qRT-PCR; these results are shown in Figure 5—figure supplement 1B.

- We originally presented CCR5 staining in two ways: first, by showing the percentage of each cell type that expressed CCR5; and second, by showing the median fluorescence intensity (MFI) of CCR5 staining in the CCR5+ population. We did this because some of the cell types exhibited a biphasic pattern of CCR5 expression, such that presenting the MFI of the entire population would not be informative. However, we recognize that this caused confusion, and so in the revised Figure 5 we show only the MFI of the CCR5+ population for each cell type. We believe this accurately and concisely demonstrates the point the there is an increase in CCR5 expression on macrophages in residual tumors.

- As described in Point #5 above, we recognize that CCR5 expression on tumor cells (and, to a lesser extent, fibroblasts) may also mediate some of the effects of CCL5. We have added text highlighting this point to the Discussion.

8) The text describing the collagen deposition in the control (GFP) tumors as compared to the CCL5-expressing recurrent tumors in Figure 6F (subsection “Macrophages express and secrete collagen and collagen deposition factors”) is a bit controversial. Shouldn't abundant collagen deposition be seen also in the recurrent control tumors if they are due to CCL5-attracted TAMS as shown in Figures 2D and 5A? One would assume that CCL5-overexpressing tumors just have more.

We agree with the reviewers’ point. We have amended the text (Discussion) to make clear that CCL5-expressing tumors have more collage deposition than control recurrent tumors, but that overall collagen deposition is high in both cohorts for the reason articulated by the reviewers.

9) The authors statement that "CCL5 had not previously been implicated in residual cell survival and recurrence" is not entirely accurate. E.g. Araujo et al. (Sci. Rep 2018; PMID:29559701) showed that in triple negative breast cancer tumor infiltrating lymphocyte recruitment was correlated with residual tumor size after neoadjuvant chemotherapy (NAC) and CCL5 expression. Moreover, Hudson et al. (Front. Oncol. 2018; PMID: 30151353) have recently shown a correlation between glioblastoma recurrence and CCL5 up-regulation in the post-treatment tumors. This statement should be rephrased accordingly.

We apologize for the oversight. We have rephrased this section per the reviewers’ suggestion and added the relevant citations (Discussion).